# A saturation hypothesis to explain both enhanced and impaired learning with enhanced plasticity

TD Barbara Nguyen-Vu[1,2*†], Grace Q Zhao[1†], Subhaneil Lahiri[3†], Rhea R Kimpo[1], Hanmi Lee[1], Surya Ganguli[1,3], Carla J Shatz[1,4], Jennifer L Raymond[1*]

[1]Department of Neurobiology, Stanford School of Medicine, Stanford, United States; [2]Department of Molecular and Cellular Physiology, Stanford School of Medicine, Stanford, United States; [3]Department of Applied Physics, Stanford University, Stanford, United States; [4]Department of Biology, Stanford University, Stanford, United States

**Abstract** Across many studies, animals with enhanced synaptic plasticity exhibit *either* enhanced or impaired learning, raising a conceptual puzzle: how enhanced plasticity can yield opposite learning outcomes? Here, we show that the recent history of experience can determine whether mice with enhanced plasticity exhibit enhanced or impaired learning in response to the same training. Mice with enhanced cerebellar LTD, due to double knockout (DKO) of MHCI H2-$K^b$/H2-$D^b$ ($K^b D^{b-/-}$), exhibited oculomotor learning deficits. However, the same mice exhibited enhanced learning after appropriate pre-training. Theoretical analysis revealed that synapses with history-dependent learning rules could recapitulate the data, and suggested that saturation may be a key factor limiting the ability of enhanced plasticity to enhance learning. Optogenetic stimulation designed to saturate LTD produced the same impairment in WT as observed in DKO mice. Overall, our results suggest that the recent history of activity and the threshold for synaptic plasticity conspire to effect divergent learning outcomes.

*For correspondence: ngbabs@ gmail.com (TDBN-V); jenr@ stanford.edu (JLR)

†These authors contributed equally to this work

## Introduction

The prospect that learning might be enhanced by enhancing synaptic plasticity has captured the imagination of many, fueled by reports of super cognition in transgenic mice with enhanced synaptic plasticity (*McConnell et al. 2009*; *Tang et al., 1999*; *Lee and Silva, 2009*; *Huh et al., 2000*; *Ito, 2002*). Synaptic plasticity is a fundamental property of neural circuits, hence its enhancement has the potential to enhance a wide range of brain functions, and benefit a wide range of patients. It could accelerate the recovery of sensory, motor, or cognitive function after a stroke or other injury; counter the decline of learning and memory in the elderly; or be used in conjunction with behavioral therapy to enhance drug rehabilitation, treatment for post-traumatic stress disorder, speech therapy, or other kinds of rehabilitation. Rapid progress in our understanding of plasticity at the synaptic level is providing targets for drug development and other molecular strategies for enhancing synaptic plasticity (*Lee and Silva, 2009*). However, optimism for this approach has been tempered by the observation that the enhancement of synaptic plasticity can, in some cases, impair learning (*Migaud et al., 1998*; *Uetani et al., 2000*; *Hayashi et al., 2004*; *Cox et al., 2003*). The finding that manipulations to enhance synaptic plasticity can either enhance or impair learning has been reported for different brain regions, and for both associative LTP and LTD (*Migaud et al., 1998*; *Uetani et al., 2000*; *Hayashi et al., 2004*; *Cox et al., 2003*; *Takeuchi et al., 2008*; *Koekkoek et al., 2005*; *McConnell et al. 2009*). Yet, there have been no experimental tests of *why* enhanced synaptic

**eLife digest** All animals can learn from their experiences. One of the main ideas for how learning occurs is that it involves changes in the strength of the connections between neurons, known as synapses. The ability of synapses to become stronger or weaker is referred to as synaptic plasticity. High levels of synaptic plasticity are generally thought to be good for learning, while low levels of synaptic plasticity make learning more difficult.

Nevertheless, studies have also reported that high levels of synaptic plasticity can sometimes impair learning. To explain these mixed results, Nguyen-Vu, Zhao, Lahiri et al. studied mice that had been genetically modified to show greater synaptic plasticity than normal mice. The same individual mutant animals were sometimes less able to learn an eye-movement task than unmodified mice, and at other times better able to learn exactly the same task. The main factor that determined how well the mice could learn was what the mice had experienced shortly before they began the training.

Nguyen-Vu et al. propose that some experiences change the strength of synapses so much that they temporarily prevent those synapses from undergoing any further changes. Animals with these "saturated" synapses will struggle to learn a new task, even if their brains are normally capable of high levels of synaptic plasticity. Notably, even normal activity appears to be able to put the synapses of the mutant mice into a saturated state, whereas this saturation would only occur in normal mice under a restricted set of circumstances. Consistent with this idea, Nguyen-Vu et al. showed that a specific type of pre-training that desaturates synapses improved the ability of the modified mice to learn the eye-movement task. Conversely, a different procedure that is known to saturate synapses impaired the learning ability of the unmodified mice.

A future challenge is to test these predictions experimentally by measuring changes in synaptic plasticity directly, both in brain slices and in living animals. The results could ultimately help to develop treatments that improve the ability to learn and so could provide benefits to a wide range of individuals, including people who have suffered a brain injury or stroke.

plasticity can have these opposite effects at the behavioral level. Moreover, despite extensive theoretical study of how enhanced plasticity can impair *memory* (the plasticity-stability dilemma; *Toulouse et al., 1986*; *Carpenter and Grossberg, 1987*; *Amit and Fusi, 1994*; *Amit and Fusi, 1992*; *Gerrow and Triller, 2010*; *Frey and Morris, 1997*; *Reymann and Frey, 2007*; *Clopath et al., 2008*; *Barrett et al., 2009*; *Redondo and Morris, 2011*), there has been little theoretical treatment of how enhanced plasticity could impair *learning.* Hence, the principles governing the learning outcome under conditions of enhanced plasticity have remained elusive, as have the principles for promoting enhanced learning. This fundamental gap in our understanding of how enhanced synaptic plasticity functions in the context of an intact neural circuit is limiting the application of synaptic plasticity enhancers in patients who could potentially benefit from this approach. We combined experiment and theory to address this conceptual gap.

We measured learning in mice deficient in molecules of the class-I major histocompatibility molecule (MHCI) complex, which have enhanced synaptic plasticity in multiple brain regions, including the cortex, hippocampus, thalamus, and cerebellum (*Huh et al., 2000*; *Syken et al., 2006*; *McConnell et al. 2009*; *Lee et al., 2014*). We focused on the cerebellum to take advantage of the known links between synaptic plasticity and cerebellum-dependent oculomotor learning. In the cerebellum, classical MHCI H2-K$^b$ (*H2-K$^b$*) and MHCI H2-D$^b$ (*H2-D$^b$*) are highly expressed in Purkinje cells, and double-knockout mice, MHCI *H2-K$^b$/H2-D$^{b-/-}$* (*K$^b$D$^{b-/-}$*; *Vugmeyster et al., 1998*; *Schott et al., 2003*; referred to as double knockout (DKO) here), exhibit enhanced associative LTD at the parallel fiber-Purkinje cell synapses (pf-Pk LTD) (*McConnell et al. 2009*). For many years, pf-Pk LTD was widely considered to be the mechanism of all cerebellum-dependent learning (*Ito, 2002*); however, recent evidence from animals with disrupted pf-Pk LTD suggests it contributes selectively to certain forms of cerebellum-dependent learning and not others (*Boyden et al., 2006*; *Hansel et al., 2006*; *Titley et al., 2010*; *Schonewille et al., 2011*; *Aiba et al., 1994*; *Shibuki et al., 1996*; *Endo et al., 2009*; *Feil et al., 2003*; *Lee et al., 2009*; *Li et al., 1995*; *Miyata et al., 2001*).

We leveraged a set of closely related oculomotor learning tasks with different dependence on pf-Pk LTD to analyze how enhanced pf-Pk LTD functions in an intact circuit.

## Results

### Impaired motor learning in mice with enhanced cerebellar LTD

We tested the ability of DKO mice to adaptively modify their vestibulo-ocular reflex (VOR). The VOR is an eye movement response to a vestibular stimulus, which functions to stabilize images on the retina during head motion. In wild type mice, motor learning can adaptively increase or decrease the amplitude of the VOR to improve image stabilization. Previous work has suggested that LTD contributes selectively to VOR learning when training to increase the amplitude of the VOR is done using high-frequency ($\geq$1 Hz) visual-vestibular stimuli, and much less so, if at all, when VOR learning is tested with other training paradigms (*Boyden et al., 2006*; *Hansel et al., 2006*; *Titley et al., 2010*; *Schonewille et al., 2011*; *Aiba et al., 1994*; *Shibuki et al., 1996*; *Endo et al., 2009*; *Feil et al., 2003*; *Lee et al., 2009*; *Li et al., 1995*; *Miyata et al., 2001*). We found that mice with enhanced pf-Pk LTD exhibited the same, specific VOR learning deficit. DKO mice were significantly impaired in learning to increase the amplitude of the VOR when training was done using 1 Hz visual-vestibular stimuli (*Figure 1D*, *solid bars*; *Figure 1—source data 1*). However, as previously reported in LTD-impaired mice (*Boyden et al., 2006*; *Schonewille et al., 2011*), there was no significant impairment of learning to increase the VOR when training was done with lower frequency visual-vestibular stimuli of 0.6 Hz (*Figure 1—figure supplement 1*, *top*), or when learning to decrease the VOR at either training frequency (*Figure 1E*, *solid bars*, *Figure 1—figure supplement 1*, *bottom*). Baseline performance of the VOR and visually-driven oculomotor behaviors were indistinguishable between DKO and WT mice (*Figure 1—figure supplements 2–3*), suggesting an impairment of the learning mechanism itself, rather than other sensory or motor deficits.

The impaired learning phenotype in DKO mice could be attributed to the loss of H2-D$^b$ expression in Purkinje cells, the post-synaptic site of pf-Pk LTD. Although H2-K$^b$ and H2-D$^b$ expression is not exclusive to the cerebellar Purkinje cells, virally-mediated rescue of H2-D$^b$ expression specifically in Purkinje cells of the flocculus (*Figure 1A–C*), the cerebellar region necessary for VOR learning, of adult global DKO mice, was sufficient to rescue their impaired VOR-increase learning (*Figure 1D*, *hatched bars*). Thus, rescue of H2-D$^b$ expression in adult neurons can restore normal function, even in animals that developed in the absence of this molecule, indicating that the role of MHCI molecules is not confined to the developing nervous system, but actively regulates plasticity in adults as well. Expression of H2-D$^b$ had no effect on VOR-decrease learning (*Figure 1E*, *hatched bars*), which is insensitive to perturbations of pf-Pk LTD (*Boyden et al., 2006*; *Hansel et al., 2006*; *Schonewille et al., 2011*).

### Climbing fiber stimulation in WT mice recapitulates the DKO phenotype

The similarity of the learning deficit in mice with enhanced pf-Pk LTD to that previously reported in mice with impaired pf-Pk LTD (*Boyden et al., 2006*) suggested the possibility of a similar underlying cause. We hypothesized that in both cases, the behavioral deficit could reflect the unavailability of pf-Pk LTD during training. In particular, the lower induction threshold for pf-Pk LTD in the DKO mice (*McConnell et al. 2009*) could allow normal basal activity in the circuit to aberrantly recruit LTD and deplete the pool of LTD-eligible synapses. Thus, the capacity for LTD could be exhausted, i.e., saturated, even before the start of training, rendering the circuit unable to support new learning that depends on pf-Pk LTD (*Figure 2A*).

To assess whether the saturation of LTD could produce a motor learning phenotype like that observed in the DKO mice, we conducted stimulation experiments. In other brain areas, direct stimulation of the relevant circuits to induce and saturate plasticity has been shown to occlude or impair subsequent learning. We used a similar approach to test whether saturation of LTD can produce the selective impairment of high-frequency VOR-increase learning observed in the DKO mice. Climbing-fiber stimulation is known to induce LTD in simultaneously active pf-Pk synapses (*Crepel and Jaillard, 1991*; *Ekerot and Kano, 1985*; *Ito and Kano, 1982*). Therefore, we optogenetically stimulated the climbing fiber input to the cerebellar flocculus to elevate the level of pf-Pk LTD in WT mice prior to VOR training (*Figure 2A*, *cyan arrow*).

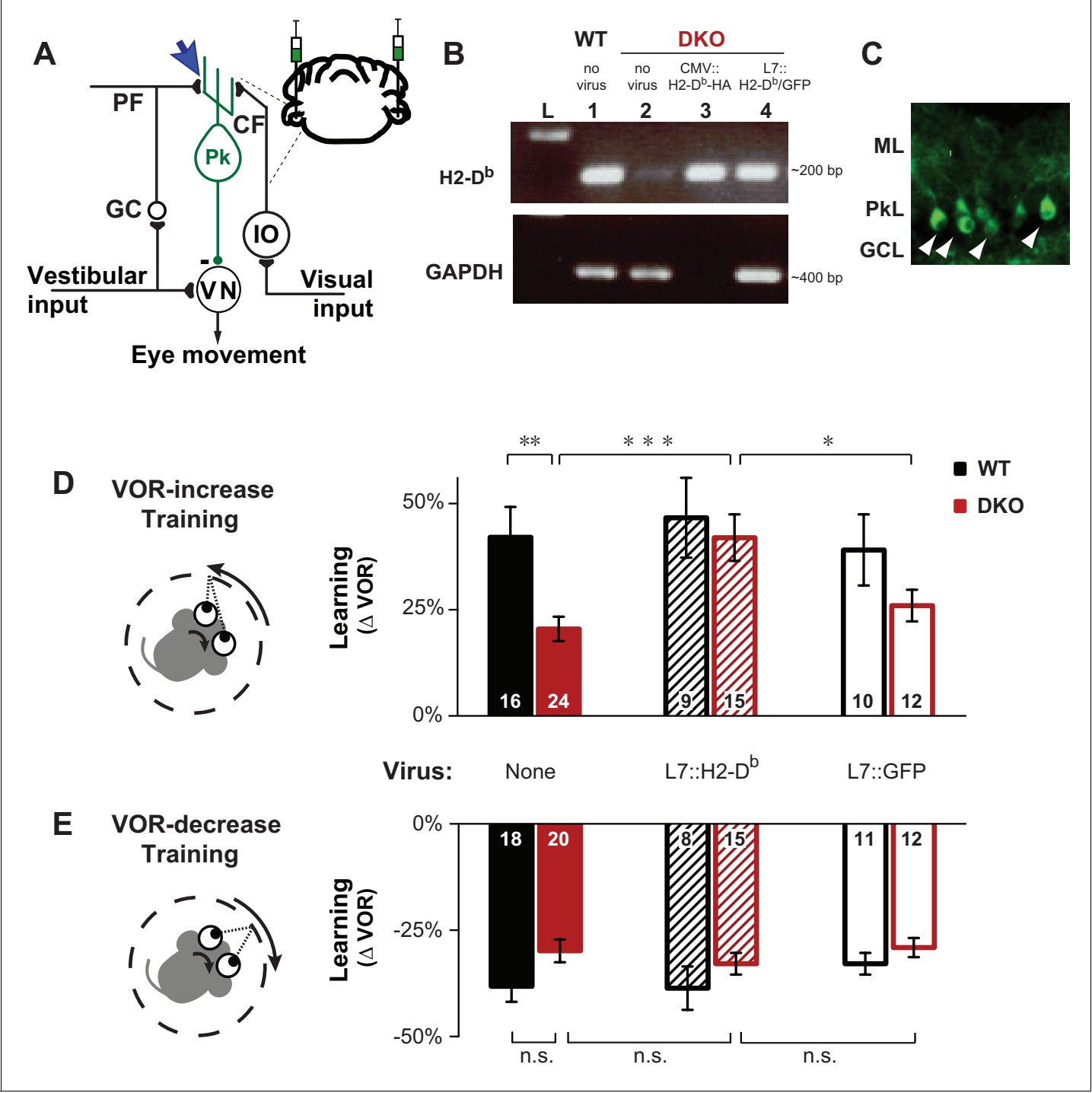

**Figure 1.** Rescue of H2-D<sup>b</sup> expression in adult Purkinje cells rescues learning impairment in DKO mice with enhanced cerebellar plasticity. (**A**) Circuit for VOR learning. Vestibular input drives eye movements via a direct pathway through the vestibular nuclei (VN), and a side-loop through the granule cells (GC), parallel fibers (PF), and Purkinje cells (Pk) of the cerebellar flocculus. The climbing fiber (CF) input to the Purkinje cells from the inferior olive (IO) carries visual signals, and can trigger LTD in the parallel fiber-to-Purkinje cell synapses (*blue arrow*), which is enhanced in mice deficient in the Class-I major histocompatibility molecules H2-K$^b$ and H2-D$^b$ ($K^bD^{b-/-}$; referred to as double knockout, DKO). A lentiviral construct expressing H2-D$^b$ under the control of the Purkinje cell-specific L7 promoter was injected bilaterally into the flocculi of adult mice (see Materials and methods for details). (**B**) RT-PCR confirmed the presence of H2-D$^b$ mRNA in the cerebellar flocculus of DKO mice injected with the L7::H2-D$^b$-T2A-GFP virus (lane 4). Lane 1: Positive control, WT (thalamus); Lane 2: Negative control, DKO (spleen); Lane 3: DKO (flocculus) infected with CMV::H2-D$^b$-HA; Lane 4: DKO (flocculus) infected with L7::H2-D$^b$-T2A-GFP. Ladder in the left lane. (Lane 3, CMV::H2-D$^b$-HA, is a positive control for detection of H2-D$^b$ expression in cerebellum, but because it was not restricted to Purkinje cells, it was not used further in this study; GAPDH was not loaded; full details in Materials and

*Figure 1 continued on next page*

*Figure 1 continued*

methods). (C) Floccular Purkinje cells of DKO mice infected with L7::H2-D$^b$-T2A-copGFP virus (*white arrowheads*) and stained with anti-copGFP immunohistochemistry. Molecular layer (ML), Purkinje cell layer (PkL), Granule cell layer (GCL). (D) Training to increase the VOR. Left, A vestibular stimulus was paired with oppositely directed visual stimulus motion. Right, DKO mice (*solid red*) were impaired on VOR-increase learning compared to wild type mice (WT; *solid black*) (**p=0.004, t$_{(38)}$ = 3.08). Virally-mediated expression of H2-D$^b$ in Purkinje cells of the adult cerebellar flocculi (L7::H2–D$^b$) rescued the learning deficit in DKO mice (*hatched red*; ***p=0.0005, t$_{(37)}$ = 3.81 vs DKO without virus, *solid red*), so that they learned as well as WT mice injected with the same virus (*hatched black*; n.s. p=0.65, t$_{(22)}$ = 0.46) and better than DKO mice that received virus expressing only GFP (L7::GFP, *open red*, *p=0.03, t$_{(25)}$ = 2.290). Virally-mediated expression of H2-D$^b$ had no significant effect on learning in the WT mice (*hatched vs. solid black*, n.s. p=0.70, t$_{(23)}$ = 0.39), and expression of GFP had no effect in DKO mice (*open vs solid red*; n.s. p=0.26, t$_{(34)}$ = 1.13) or WT mice (*open vs. solid black*; n.s. p=0.79, t$_{(24)}$ = 0.26). Mean ± s.e.m. In this and all figures, *numbers in bars* indicate n = number of animals. (E) Training to decrease the VOR. Left, A vestibular stimulus was paired with a visual stimulus that moved with the head. Right, VOR-decrease learning in DKO mice (*solid red*) was not significantly different from WT (*solid black*) (n.s. p=0.08, t$_{(36)}$ = 1.86). Expression of H2-D$^b$ had no significant effect on VOR-decrease learning in DKO mice (*hatched vs. solid red*; n.s. p=0.43, t$_{(33)}$ = 0.79), and was not different from mice that received control virus expressing only GFP (*hatched vs. open red*; n.s. p=0.29, t$_{(25)}$ = 1.08). Mean ± s.e.m.

The following source data and figure supplements are available for figure 1:

**Source data 1.** Rescue of H2-D$^b$ expression in adult Purkinje cells rescue learning impairment in DKO mice with enhanced cerebellar plasticity.
**Figure supplement 1.** DKO mice were selectively impaired on high-frequency VOR-increase learning.
**Figure supplement 2.** Baseline oculomotor performance was normal in DKO mice.
**Figure supplement 3.** Control for eye movements during training.

Climbing fibers were optogenetically stimulated for 30 min (250 ms trains of three 2 ms light pulses, repeated every 1 s) while the mouse was head-restrained in the dark without visual or vestibular stimuli. During normal VOR learning, climbing fiber activation is thought to induce LTD selectively in those pf-Pk synapses activated by the visual and vestibular stimuli used to induce learning. In contrast, optogenetic climbing fiber stimulation delivered in the absence of such stimuli should induce LTD randomly in spontaneously active pf-Pk synapses. This non-specific LTD-induction procedure did not affect the amplitude of the VOR, measured after climbing fiber stimulation (*Figure 2—figure supplement 1B*). This is consistent with the normal baseline VOR amplitude in DKO mice (*Figure 1—figure supplement 2A*), and in wild type mice after lesions of the flocculus (*Rambold et al., 2002*; *Koekkoek et al., 1997*; *Katoh et al., 2005*). Together, these observations indicate that non-specific manipulations of the flocculus are not sufficient to have a coordinated effect on the VOR behavior, and that LTD only increases the amplitude of the VOR if it is induced selectively in the appropriate subset of pf-Pk synapses. Nevertheless, if non-specific LTD depleted the pool of LTD-eligible synapses, it could impair subsequent LTD-dependent learning (*Figure 2—figure supplement 2*). Accordingly, VOR-increase learning was impaired after climbing fiber stimulation (*Figure 2B*, CF Stim, cyan vs. Sham Stim, black; *Figure 2—source data 1*). Sham stimulation controls exhibited VOR-increase learning, confirming that disrupted VOR-increase learning was specific to stimulation of climbing fibers, rather than reflecting nonspecific, optical or mechanical perturbation of the circuit. Notably, climbing fiber stimulation before training did not perturb subsequent VOR-decrease learning, which also relies on the cerebellar flocculus (*Koekkoek et al., 1997*; *Rambold et al., 2002*) but is insensitive to disruptions of pf-Pk LTD (*Boyden et al., 2006*) (*Figure 2B*, dashed traces). The specificity of the learning impairment in WT mice after climbing fiber stimulation indicates that elevated levels of pf-Pk LTD prior to training can produce a phenotype like that observed in the DKO mice (*Figure 1D*).

## Behavioral pre-training converts impaired learning of DKO mice to enhanced learning

If elevated pf-Pk LTD prior to training is contributing to the learning impairment in the DKO mice, then any procedure that reverses pf-Pk LTD might reset the synapses to a state more capable of supporting LTD-dependent learning. Pf-Pk LTD can be actively reversed by post-synaptic LTP of the same synapses (*Lev-Ram et al., 2003*), providing a cellular mechanism for reversing LTD saturation.

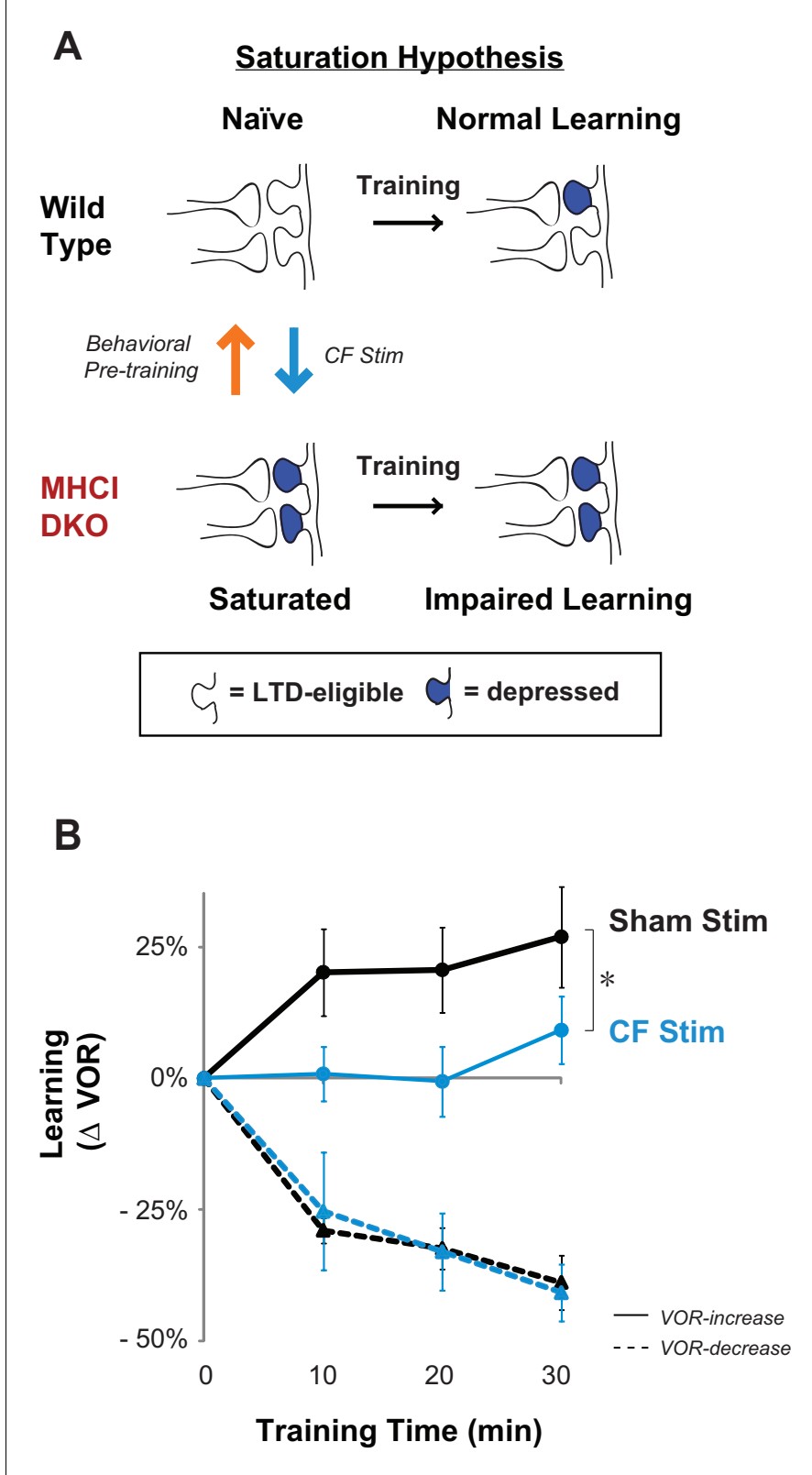

**Figure 2.** Elevated LTD before training impairs LTD-dependent learning. (**A**) Saturation hypothesis to explain impaired learning with enhanced synaptic plasticity. **Top,** In naïve WT mice, at the start of training, synapses are presumably available (*white synaptic spines*) to selectively undergo associative synaptic plasticity (long-term depression, LTD; *blue spines*) during training, thereby supporting normal learning. **Bottom,** In DKO mice, the

*Figure 2 continued*

lower induction threshold for LTD could enable spontaneous activity in the circuit to aberrantly recruit LTD at a random subset of spontaneously active synapses before training, thereby depleting the pool of synapses eligible to undergo LTD, and preventing normal learning. Behavioral pre-training (*orange arrow*) restores the capacity for LTD-dependent learning in the DKO mice (*Figure 3*). We tested whether LTD saturation and impairment of LTD-dependent learning can be induced in WT mice with climbing fiber stimulation (*cyan arrow*; *Figure 2B*). (**B**) Climbing fiber stimulation in WT mice before VOR training recapitulates the learning impairment in the DKO mice. Optogenetic stimulation of climbing fibers for 30 min, to induce pf-Pk LTD in the flocculus of WT mice, blocked subsequent VOR-increase learning (*solid cyan trace*; *p=0.03, $F_{(1,10)}$ = 5.912, two-factor repeated measures ANOVA, CF stim n = 6, Sham n = 6) but had no effect on VOR-decrease learning (*dashed cyan trace;* n.s. p=0.68, $F_{(1,5)}$ = 0.20) relative to sham stimulation controls in animals that did not express ChR2 in the climbing fibers (*black*). Mean ± s.e.m.

The following source data and figure supplements are available for figure 2:

**Source data 1.** Elevated LTD before training impairs LTD-dependent learning.

**Figure supplement 1.** Climbing fiber stimulation did not permanently impair VOR-increase learning.

**Figure supplement 2.** Non-specific LTD may have no immediate effect on behavior, yet deplete the pool of synapses available to support LTD-dependent learning.

Moreover, behavioral VOR-decrease training has been shown to rapidly reverse any evidence of prior VOR-increase learning (*Boyden and Raymond, 2003*), suggesting that it rapidly reverses any pf-Pk LTD or other plasticity induced during VOR-increase learning. Therefore, we tested whether VOR-decrease training could put the VOR circuit of DKO mice into a state more capable of supporting LTD-dependent VOR-increase training.

Mice were given thirty minutes of VOR-decrease pre-training immediately before VOR-increase training. In the DKO mice, this pre-training significantly enhanced subsequent VOR-increase learning relative to DKO mice that did not receive the pre-training (*Figure 3A*, *compare red traces and bars in middle vs. left panels*; *Figure 3—source data 1*). Notably, pre-training had the opposite effect on WT mice, impairing subsequent VOR-increase learning (*Figure 3A*, *compare black traces in middle vs. left panels*). Since the pre-training enhanced learning in the DKO mice, but impaired learning in the WT mice, the DKO mice learned better than the WT after pre-training. Hence, the pre-training not only reversed the learning impairment, but also revealed a capacity for enhanced learning in the DKO mice relative to WT mice (*Figure 3A*, *compare black vs. red traces in middle panel*).

The enhanced learning in the DKO mice relative to WT mice after pre-training could not be explained by differences in the efficacy of VOR-decrease learning (*Figure 3—figure supplement 1*), nor could it be explained by more rapid forgetting of the effects of pre-training in the DKO mice, because VOR-decrease learning was retained normally (*Figure 3—figure supplement 2*). Also, the enhanced learning was unmasked specifically by pre-training with the associative VOR-decrease training paradigm, which paired visual and vestibular stimuli. Simply decreasing the VOR amplitude with a non-associative, habituation paradigm, which presented the vestibular stimulus alone, was not sufficient to unmask the enhanced learning (*Figure 3A*, *right, Vestibular Only Pre-training*). Thus, only the appropriate pre-training experience can put the circuit of the DKO mice into a state that enables their enhanced synaptic plasticity to support enhanced learning (*Figure 2A*, *orange arrow*).

The enhanced learning phenotype of the DKO mice had features in common with the impaired learning phenotype in these mice. First, the enhanced learning phenotype, like the impaired learning phenotype (*Figure 1D*, *solid bars*, *Figure 1—figure supplement 1*, *top*), was only observed when training was done using high-frequency (1.0 Hz) visual-vestibular stimuli (*Figure 3A*, *middle panel*), but not a slightly lower stimulus frequency of 0.6 Hz (*Figure 3—figure supplement 3*). In addition, the enhanced learning phenotype, like the impaired learning phenotype in the DKO mice, reverted to WT phenotype after virally-mediated expression of H2-D[b] in the Purkinje cells of adult DKO mice (compare *Figure 3B*, *left panel,* with *Figure 3A*, *middle panel*, and *Figure 1D*, *hatched bars*). The commonality of these features suggests that both the enhanced and impaired learning phenotypes of DKO mice share a common mechanism, involving the same set of synapses.

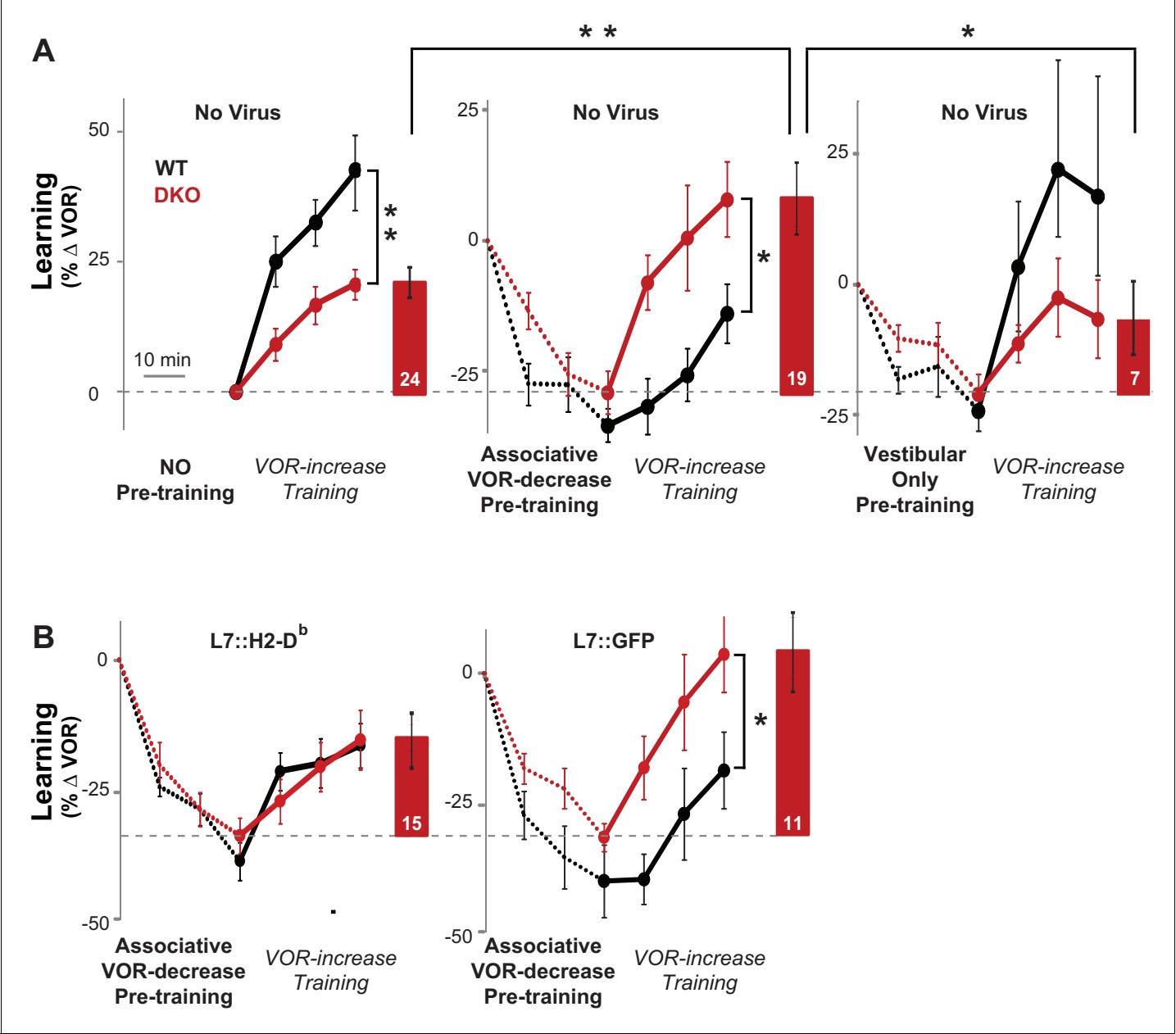

**Figure 3.** Behavioral pre-training reveals enhanced learning in mice with enhanced LTD. (**A**) The same VOR-increase training procedure induced dramatically different learning outcomes in the DKO mice with different pre-training procedures (p=0.01, F = 5.153, ANOVA). *Left,* Without pre-training, DKO mice with enhanced pf-Pk LTD were impaired on VOR-increase learning (**p=0.002, $F_{(1,38)}$ = 11.08, two-factor repeated measures ANOVA; WT n = 16,. DKO n = 24,). *Middle,* Pre-training with an associative VOR-decrease paradigm that was not significantly different between the genotypes (*dotted lines, p=0.19, $F_{(1,29)}$ = 1.79; WT n = 12, DKO n = 19) reversed the learning impairment in DKO mice (*red*) so that they learned more than WT (*black*) during subsequent VOR-increase training (*p=0.02, $F_{(1,29)}$ = 5.95; WT n = 12, DKO n = 19). *Right,* Pre-training with a vestibular stimulus alone decreased the VOR gain comparably between the two genotypes (*dotted line, p=0.30, $F_{(1,17)}$ = 1.25; WT n = 6, DKO n = 7), but there was no improvement of subsequent VOR-increase learning in the DKO mice relative to WT mice (p=0.13, $F_{(1,11)}$ = 2.70; WT n = 6, DKO n = 7). In DKO mice, VOR-increase learning was better after associative VOR-decrease pre-training compared with no pre-training (**p=0.005, Fischer's LSD) or vestibular-only pre-training (*p=0.03) (*compare red bar graphs and learning curves*). In contrast, in WT mice, VOR-increase learning was *worse* after associative VOR-decrease pre-training compared with no pre-training (*p=0.037, Fischer's LSD) or vestibular only pre-training (*p=0.049) (*compare black learning curves*). Learning is plotted on the same scale in each plot, and aligned on the values at the start of VOR-increase training for DKO mice. Mean ± s.e. m. (**B**) Virally-mediated rescue of H2-D$^b$ expression in floccular Purkinje cells (L7::H2-D$^b$, *left*) eliminated the enhanced VOR-increase learning in DKO mice after associative VOR-decrease pre-training (*compare with middle panel of **A***), so that learning was indistinguishable from WT mice injected with the same virus (*VOR-increase learning, p=0.98, $F_{(1,22)}$ = 0.0004; VOR-decrease pre-training, p=0.53, $F_{(1,22)}$ = 0.40; two-factor repeated measure ANOVA;* *Figure 3 continued on next page*

*Figure 3 continued*

WT n = 9; DKO n = 15). The enhanced VOR-increase learning phenotype was present in DKO mice that received control virus expressing only GFP (L7:: GFP, *right,* p=0.05, $F_{(1,18)}$ = 4.29; WT n = 9, DKO n = 11) although the VOR-decrease pre-training itself was not significantly different between the two genotypes (p=0.20, $F_{(1,18)}$ = 1.75; WT n = 9, DKO n = 11). Mean ± s.e.m.
The following source data and figure supplements are available for figure 3:

**Source data 1.** Behavioral pre-training reveals enhanced learning in mice with enhanced LTD.
**Figure supplement 1.** Control for efficacy of VOR-decrease pre-training.
**Figure supplement 2.** Normal retention of VOR-decrease learning in DKO mice.
**Figure supplement 3.** No enhanced learning phenotype was observed in DKO mice when tested using lower visual-vestibular stimulus frequencies.

## Modeling predicts strong saturation of LTD and difficult to reverse synaptic states

The experimental results above raise the possibility that the learning impairment of the DKO mice relative to WT mice could result from the saturation of LTD. We conducted computational modeling studies to further assess the plausibility of this hypothesis, and, more generally, to develop an understanding of the properties of synaptic plasticity that can support *both* enhanced *and* impaired learning, depending on recent experience. First, we examined how the competition between two opposing factors determines the learning outcome. One factor is the enhanced intrinsic propensity for synapses to undergo LTD, which alone would enhance learning. The other factor is the depletion of the number of synapses eligible for LTD, which alone would impair learning. Second, we characterized the properties of synaptic models that could reproduce the opposite effects of behavioral pre-training, namely impairing learning in WT mice, and enhancing learning in DKO mice. The modeling generates predictions about the essential features of synapses that could support both the impaired and enhanced learning outcomes we observed empirically. Moreover, it provides general insights about how the prior history of activity in the circuit interacts with the threshold for plasticity to determine whether learning is impaired or enhanced.

We adopted a theoretical framework (*Fusi et al., 2005*; *Fusi and Abbott, 2007a*; *Lahiri and Ganguli, 2013*) previously used to study memory, to compare different synaptic models. Each model incorporated three key experimental findings: (1) the selective contribution of pf-Pk LTD to VOR-increase and not VOR-decrease learning (*Boyden et al., 2006*); (2) the observation that pf-Pk LTD is easier to induce in the DKO mice (i.e., the 'lower threshold' for LTD induction *McConnell et al. 2009*); and (3) the ability of VOR-decrease training to reverse the effects of VOR-increase training (*Boyden and Raymond, 2003*).

The contribution of pf-Pk LTD to VOR-increase learning was modeled by using the rate at which synapses transitioned to a depressed state during training as the measure of VOR-increase learning (*Ito, 1972*). Training to increase the VOR was modeled by increasing the rate of LTD 'events', which can be thought of as the rate at which patterns of neural activity occur with the potential to induce LTD (*see Appendix for details*). The lower threshold for pf-Pk LTD in the DKO mice was modeled as an increase in the probability that an LTD 'event' (i.e., near-coincident parallel fiber and climbing fiber activation) will actually induce LTD, which we will refer to as the 'intrinsic' LTD rate.

In vitro, Pf-Pk LTD can be actively reversed by post-synaptic LTP of the same synapses (*Lev-Ram et al., 2003*). Therefore, the ability of VOR-decrease training to reverse the effects of VOR-increase training was modeled by increasing the rate of LTP events at the pf-Pk synapses during VOR-decrease training (*Boyden and Raymond, 2003*). However, we note that the mechanism of VOR-decrease learning is not currently known. Moreover, known asymmetries in VOR-increase and VOR-decrease learning (*Boyden and Raymond, 2003*; *Kimpo et al., 2005*) suggest that the mechanism of VOR decrease learning is not simply pf-Pk LTP. Therefore, we do not attempt to model the decrease in VOR gain itself, but to merely capture the effects of VOR-decrease training on the pf-Pk synapses, and, more specifically, its effect on the availability of pf-Pk synapses to undergo LTD during VOR-increase learning.

We implemented synaptic models with different numbers of potentiated and depressed states and different probabilities of transitioning between states (*Montgomery and Madison, 2002*; *Petersen et al., 1998*), and compared their ability to reproduce our empirical observations of VOR learning in wild type and DKO mice (*Figure 4A*). Specific synaptic models were considered for their analytical tractability and prevalence in theoretical treatment. In all of the models, the lower threshold for LTD in the DKO mice interacted with the rate of spontaneous LTD events caused by basal activity in the circuit (basal rate of parallel fiber and climbing fiber coactivation) to bias the initial distribution of synapses towards the depressed state(s) prior to learning (*Figure 4B,D*, *top right, blue bars*). Neurons in the depressed states were ineligible or less eligible to undergo additional LTD. Thus, for DKO mice, the outcome of VOR-increase training depended upon a competition between two opposing forces: (1) an enhanced intrinsic propensity for eligible synapses to undergo LTD, which alone would enhance learning; and (2) depletion of synapses eligible for LTD, i.e., saturation of LTD, which alone would impair learning.

Classical models of LTP and LTD did not reproduce our observation of impaired learning with enhanced plasticity. We tested a simple binary synapse model (*Figure 4B*), and a more generalized linear multistate model with multiple synaptic strengths (*see Appendix*). These models encapsulate classical notions of synaptic plasticity as straightforward changes in synaptic strength, to a maximal or minimal bound. However, one can show mathematically, that for all values of the parameters of these models, the enhanced intrinsic LTD rate dominates the saturation effect, at least for the initial phase of learning. Thus, these models incorrectly predict that enhanced plasticity would enhance VOR-increase learning (*Figure 4B*, *solid red vs. black trace, green bracket; see Appendix for an analytical solution and predictions of models for longer time scales*). Thus, classical models of synaptic plasticity could not readily account for the behavioral results observed empirically.

To predict impaired learning with enhanced plasticity, a mechanism to amplify the effect of depleting the synapses eligible for LTD was required. We first considered a synaptic model in which LTD driven by spontaneous activity in the circuit would not only deplete the LTD-eligible pool, but also retard LTD in the remaining LTD-eligible synapses (*Figure 4C*), as one might expect, for example, if a protein necessary for LTD induction was present in a cell in limited quantities. This resource-depletion model reproduced the impaired learning of DKO mice, however, it failed to predict the empirical observation of impaired learning after pre-training in WT mice (*Figure 4C*, *dotted black vs. solid black trace, green bracket*). To account for this latter observation, the synaptic architecture had to include 'stubborn' synaptic states whereby too many LTD-reversing events can impair the capacity for subsequent LTD.

One model that possesses both essential properties of amplified saturation effects and stubborn synaptic states is a serial model (*Leibold and Kempter 2008*; *Ben Dayan Rubin and Fusi, 2007*) with only two different synaptic weights, but with each weight associated with multiple, internal states (*Figure 4D*). Enhancing LTD in these models leads to an exponential distribution over synaptic states, which strongly depletes the pool of synapses available to express LTD. This exponential distribution of synapses (*Figure 4D*, *top right*) can account for the impaired learning phenotype in the DKO mice by providing sufficient depletion of LTD-eligible synapses by spontaneous basal activity to overwhelm the higher intrinsic LTD rate in the remaining LTD-eligible synapses. In this model, pre-training in the DKO mice reversed this saturation bias (*Figure 4D*, *bottom right vs. top right*), allowing the higher intrinsic LTD rate to dominate. In contrast, WT mice started with many LTD-eligible synapses, but pre-training pushed synapses deep into the chain of potentiated states, thereby reducing their ability to undergo a subsequent transition to a depressed state (*Figure 4D*, *bottom left, orange bars*). Notably, the model predicts that with extended VOR-increase training, the advantage conferred on the DKO mice relative to WT mice by pre-training should disappear (*see Appendix*).

Other synaptic models in which the capacity for a synaptic weight to change depends on the history of prior plasticity events could also account for our empirical observations. Models with such metaplasticity include the cascade model (*Fusi et al., 2005*) (*see Appendix*), and a multistate model with multiple synaptic strengths and lower transition probabilities for the deeper states (*Figure 4E*). Given appropriate parameters, these models are capable of reproducing all of the qualitative learning outcomes observed experimentally (*Figure 4A*, *right*), in contrast to the classical, binary or linear multistate models of plasticity, which are unable to do so for *any* choice of parameters. This successful class of models illustrates general principles about how the enhancement of plasticity at a given

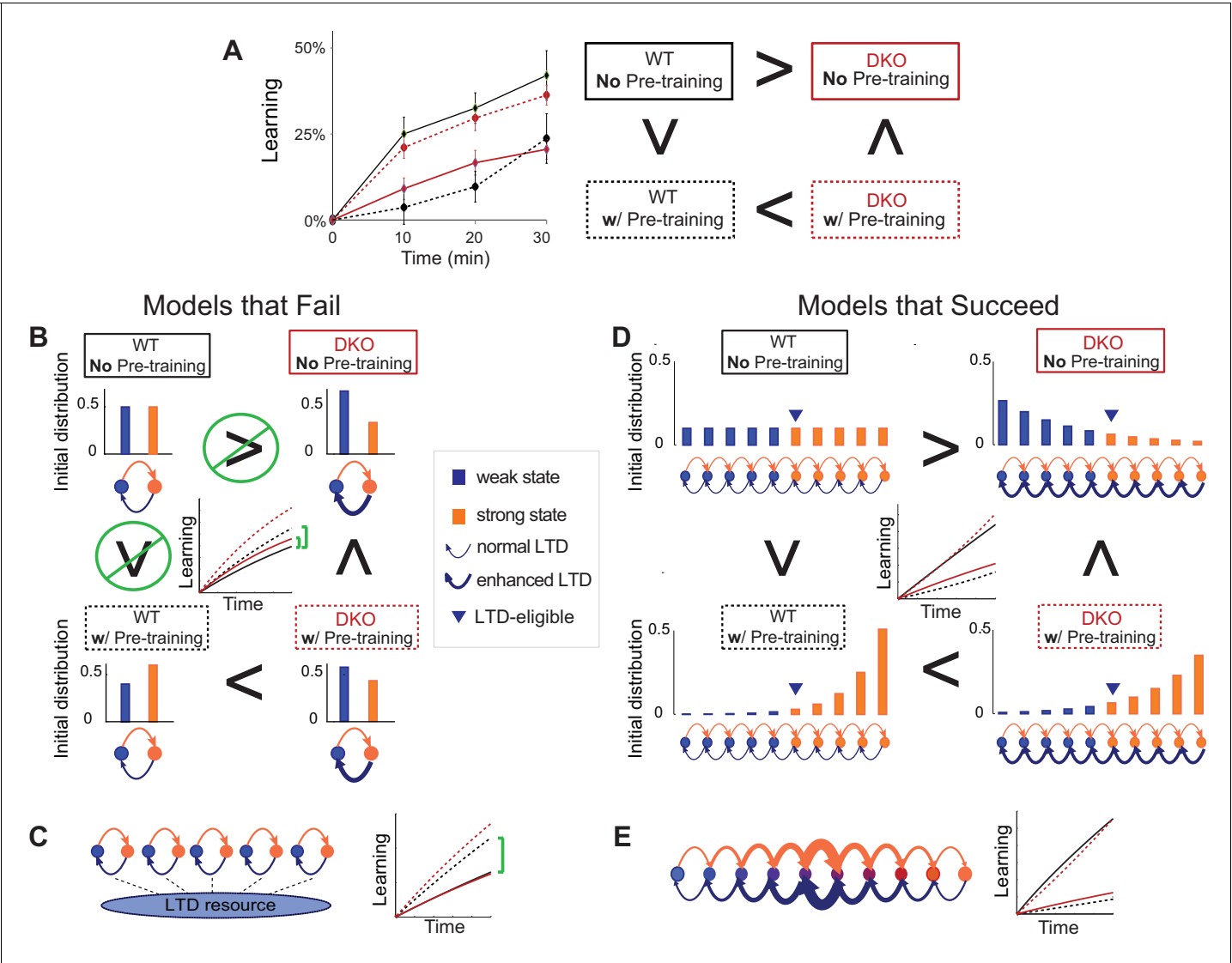

**Figure 4.** Synaptic models with amplified saturation effects and stubborn synaptic states account for learning in mice with normal and enhanced LTD. (A) Four empirical comparisons constrain the models. *Left,* Empirical results replotted from *Figure 3*, with all curves aligned to the start of VOR-increase training, P values can be found in the legend for *Figure 3*. *Right,* Less than and greater than symbols (< and >) indicate which mice exhibited greater VOR-increase learning. In all panels of *Figure 4*: red, DKO mice; *black,* WT mice; *solid lines,* no pre-training; *dashed lines,* with VOR-decrease pre-training. (B) A binary synapse model with a strong synaptic state (*orange*) and a weak state (*blue*). Synapses transition between the two states at the rate of depression (*blue curved arrow)* and potentiation (*orange curved arrow*). The fraction of synapses in each state prior to VOR-increase learning is indicated by *blue* and *orange bars.* VOR-increase learning is measured by the decrease in synaptic weights during training. For DKO mice, the rate of depression was higher than WT, reflecting the lower threshold for LTD, (*thick blue arrow*), hence a greater fraction of the synapses were in the weak state (*blue bars*) prior to any VOR training. VOR-decrease pre-training (*bottom panels*) increased the fraction of synapses in the strong, LTD-eligible state (*orange*) in both WT and DKO mice. *Center,* The binary synapse model predicts enhanced learning in DKO vs. WT mice without pre-training (*solid red vs solid black trace*) and enhanced learning in WT mice with vs. without pre-training (*dashed vs solid black trace*), in contradiction to the empirical results in A (*green brackets and green Ø*). (C) The pooled resource model. *Left,* The probability of synaptic depression varied with the level of a shared resource that was depleted by the occurrence of depression at other synapses. *Right,* This model fails to account for the impaired learning in WT mice after pre-training (*dashed black vs. solid black; green bracket*). (D) The serial synaptic model with multiple strong (*orange*) and weak (*blue*) states, but only two values of synaptic strength, can account qualitatively for the effects of enhanced LTD and pre-training on learning (compare *center panel* with A). Before training, the synapses were strongly biased towards the weak state in the DKO mice, reducing the fraction of LTD-eligible synapses (*blue arrowheads*), and impairing learning relative to WT (*solid red vs. solid black*), as observed empirically. VOR-decrease pre-training shifted the bias towards the strong states (*bottom panels*). In DKO mice, this increased the fraction of LTD-eligible synapses (*blue triangle*), and enhanced learning (*dashed red*). In WT mice, pre-training biased the synapses to be too deep into the chain of potentiated states, so that the fraction of LTD-eligible synapses was reduced (*blue triangle*) and learning impaired (*dashed black*). (E) The non-uniform multistate model. *Left,* Each state is of varying strength

*Figure 4 continued on next page*

*Figure 4 continued*

from strong (*orange*) to weak (*blue*), and the transition probabilities between states decay exponentially the further the state is from the center. *Right*, This model qualitatively reproduced all of the empirical observations of learning.

synapse can contribute to both impaired and enhanced learning, depending on the recent history of activity. In essence, under conditions where the recent activity leaves many synapses of WT mice in the labile, LTD-eligible states, enhancing plasticity tends to push the synapses out of these LTD-eligible states before training and thus impairs learning despite enhanced plasticity (*compare Figure 4D, top-left with top-right*). Under conditions where the recent activity leaves many synapses of WT mice in the 'stubborn," potentiated states, enhancing plasticity can push them into the labile, LTD-eligible states, and enhance subsequent learning (*compare Figure 4D, bottom-left with bottom-right*). A history-dependent alteration in the capacity to undergo additional plasticity has been documented experimentally at some synapses in the hippocampus (*Montgomery and Madison, 2002*; *Petersen et al., 1998*). Such history-dependence in the plasticity of cerebellar synapses, required by our model to explain our observed behavioral phenotypes, constitutes a key prediction that can be tested in future empirical investigations of the synaptic physiology.

## Discussion

Our results provide new insights about how enhanced synaptic plasticity can yield either enhanced or impaired learning, and begin to identify the factors that favor enhanced versus impaired learning when synaptic plasticity is enhanced. Although it was previously known that enhanced plasticity can have these opposite effects on behavior, these divergent results were obtained by different labs, using different learning tasks that depend on different brain regions, and different lines of mice with enhanced plasticity. Here, we established that the same individual animals with enhanced plasticity can respond to the same behavioral training with either enhanced or impaired learning, depending on the recent history of experience. Thus, the capacity for new learning is determined by a dynamic interplay between the threshold for synaptic plasticity and the recent history of activity.

Classical models of synaptic plasticity, in which an LTP event simply increases the synaptic strength and an LTD event simply decreases the synaptic strength, do not readily explain our behavioral data. We showed that in such models, enhancing plasticity led to enhanced learning, across the entire parameter space, and independent of the history of previous learning experiences. A critical, theoretical ingredient required to account for our data is a history-dependent form of synaptic plasticity (*Montgomery and Madison, 2002*; *Petersen et al., 1998*; *Fusi et al., 2005*) in which repeated LTD changes the internal state of the synapse into a less labile state. Then enhanced LTD can deplete the pool of labile synapses capable of supporting further learning, leading to impaired learning despite enhanced plasticity. Thus, the similar learning deficits in mice with enhanced pf-Pk LTD (*Figure 1D*) and impaired pf-Pk LTD (*Boyden et al., 2006*; *Hansel et al., 2006*) could reflect a similar underlying cause, namely the unavailability of pf-Pk LTD during training (*Figure 2A*).

More generally, our results suggest a new hypothesis for why enhanced plasticity can impair learning. Such impairments have generally been attributed to an *over-recruitment* of the enhanced plasticity mechanism at inappropriate synapses during training corrupting the memory trace (*Migaud et al., 1998*; *Koekkoek et al., 2005*; *Martin et al., 2000*). Our results raise the alternative possibility that enhancing the plasticity mechanism necessary for learning might lead, instead to its *under-recruitment* at appropriate synapses during learning, as a result of saturation (*Figure 2A*).

The possibility that synaptic plasticity can be saturated has long been recognized (*Martin et al., 2000*). Behavioral training paradigms that induce learning can occlude the subsequent induction of synaptic plasticity in brain slices (*Schreurs et al., 1997*; *Pascoli et al., 2012*; *Rioult-Pedotti et al., 1998*), and strong, artificial stimulation of neural activity in vivo to saturate a plasticity mechanism has been shown to impair subsequent learning in animals with normal synaptic plasticity (*Martin et al., 2000*; *Moser et al., 1998*), as we found for cerebellar climbing fiber stimulation (*Figure 2B*). Despite its consideration in these contexts, saturation has not been identified previously as a factor that could limit the ability of enhanced synaptic plasticity to enhance learning. Our results provide initial experimental evidence for this hypothesis. In particular, our results suggest that in

mice with a low threshold for associative synaptic plasticity, saturation of the plasticity and occlusion of further learning may occur, not only in response to the patterns of elevated neural activity that can induce saturation in WT animals, but also in response to the normal, ongoing, basal levels of activity in a circuit, in the absence of any training or neural stimulation (compare *Figure 1D* and *Figure 2B*). Thus, enhanced synaptic plasticity, in the form of a lower threshold for induction, can be opposed by a tendency for plasticity to saturate, which, in turn, limits the capacity for new learning.

The capacity for new learning could be decreased or increased by manipulations that altered activity in the VOR circuit for a few tens of minutes. In WT mice, 30 min of elevated climbing-fiber activity induced a state that prevented subsequent learning (*Figure 2B*, *CF Stim, solid cyan*). Recovery from this saturation also occurred over a timescale of tens of minutes; although VOR-increase learning was profoundly suppressed after climbing fiber stimulation, there was evidence for recovery of learning during the last few minutes of the 30 min training session (*Figure 2B*, *solid blue trace at 30 min*), and full recovery a few days later (*Figure 2—figure supplement 1C*). Likewise, the natural manipulation of circuit activity caused by 30 min of behavioral pre-training was apparently sufficient to reverse saturation in the DKO mice and transform their learning impairment into a learning enhancement (*Figure 3A*). Thus, if the recent neural activity is appropriately patterned, rapid recovery from saturation is possible, creating the potential for enhanced plasticity to support enhanced learning. This dependence on the recent history of activity may explain the difference between the results from DKO mice in vivo versus in vitro. In vitro, the levels of spontaneous activity are much lower than in vivo, hence any saturation that is present in vivo would rapidly decay in vitro, revealing enhancement rather than saturation of pf-Pk LTD (*McConnell et al., 2009*). The dependence of learning on the recent history of activity may also explain the observation of either impaired or enhanced learning on behavioral tasks that depend on different parts of the cerebellum (*McConnell et al., 2009*), which have different levels of spontaneous activity (*Zhou et al., 2014*), and different dependence of LTD induction on the patterns of neural activation (*Suvrathan et al., 2016*). In sum, in both DKO and WT mice, the capacity for new learning was highly dependent on the recent history of activity in the circuit over the previous tens of minutes.

Our findings reinforce the idea that synaptic plasticity and learning are not isomorphic: one cannot predict the learning outcome from the synaptic properties alone. Rather, the capacity for new learning is determined by a dynamic interplay between the threshold for synaptic plasticity and the recent history of activity. A better understanding of this interaction is of great clinical significance, with the potential to guide the treatment of a wide range of patients who could benefit from enhanced neural plasticity, such as those recovering from brain injury. Our results suggest that synaptic plasticity enhancers may be most effective if combined with strategies for controlling basal levels of neural activity. In particular, suppression of neural activity before training may prime enhanced learning under conditions of enhanced associative plasticity. In addition, our finding that the appropriate behavioral pre-training can unmask enhanced learning in mice with enhanced synaptic plasticity (*Figure 3A*) raises the possibility that behavioral therapy could provide an alternative to drugs in patients with pathologically altered synaptic plasticity (*Koekkoek et al., 2005*; *Yashiro et al., 2009*; *Baudouin et al., 2012*).

A better understanding of how the threshold for synaptic plasticity affects function has the potential to influence many areas of neuroscience. Synaptic plasticity plays a role in nearly all brain functions, from the most basic sensory processing to the highest cognitive functions, and from early development through aging (*Hübener and Bonhoeffer, 2014*; *Greenwood, 2007*; *Chen and Tonegawa, 1997*; *Meredith, 2015*). To get traction on the broad issue of how enhanced synaptic plasticity influences learning, we harnessed the analytical power of the relatively simple and well-characterized vestibular and oculomotor systems, and leveraged manipulations informed by specific knowledge about signaling and plasticity in those systems. Yet the finding that the enhancement of synaptic plasticity can result in either enhanced or impaired learning has been reported for many different brain regions, and for both associative LTP and LTD. Therefore, saturation should be considered as a factor that could limit the ability of enhanced plasticity to enhance learning in neural circuits throughout the brain. One can speculate that in each brain area, neural circuits have evolved to optimize the threshold for plasticity to delicately balance the need to prevent inappropriate inputs from triggering and saturating plasticity, while allowing the appropriate inputs to drive learning.

# Materials and methods

## Animals

All experimental procedures were approved by the Administrative Panel on Laboratory Animal Care at Stanford University under the animal care and use committee (IACUC) Protocol #9143, titled 'Vestibular and Visual Control of Eye Movements in Mice'. All mice were housed on a reversed 12 hr light/12 hr dark cycle, and experiments were conducted during the animals' dark cycle. After implant surgery for behavioral experiments, mice were single-housed in individual cages. All other mice were maintained in group housing of up to five animals per cage. MHCI *H2-K$^b$/H2-D$^{b-/-}$* (*K$^b$D$^{b-/-}$*; referred to as double knockout (DKO)) mice on a C57BL/6 genetic background (*Vugmeyster et al., 1998*; *Schott et al., 2003*) were maintained as a homozygous breeding colony. Age-matched C57BL/6 WT mice (RRID:IMSR_JAX:000664) were purchased from Jackson Laboratory.

## Surgery

### Eye coil and head post implants

Mice were surgically prepared for behavioral experiments as previously described (*Boyden and Raymond, 2003*). In brief, while under anesthesia with ketamine/dexmedetomidine followed by isoflurane, a custom-built head post was attached to the top of the skull using anchor screws and dental acrylic. A small, copper scleral search coil (IET, Marly, Switzerland), 1 mm in diameter, was implanted on the temporal side of one eye beneath the conjunctiva for use in measuring eye movements. The search-coil leads were threaded subcutaneously and soldered to a two-pin connector that was also cemented to the anchor screws with dental acrylic. Animals were allowed to recover from surgery for 4–5 days before behavioral experiments.

### Virus injections in the cerebellar flocculus

In some mice, lentivirus carrying H2-D$^b$, H2-D$^b$/GFP or GFP under the L7/pcp-2 promoter (see below) was used to drive expression specifically in cerebellar Purkinje cells. Mice were anesthetized as described above and craniotomies were made on both sides of the skull over the periotic capsule, which overlies the cerebellar flocculus. Approaching through the pinna, 1 ul of lentivirus was injected at a rate of 0.15 ul/min into each flocculus at a depth of 1250 to 1500 um, using a Harvard Apparatus 11plus pump. A minimum of two weeks following injection was allowed for viral expression before the mice were tested.

### Virus injections in the inferior olive

For optogenetic stimulation experiments, virus carrying ChR2 was injected into the dorsal cap of Kooy of the inferior olive (IO), to target ChR2 to the neurons giving rise to the climbing fiber inputs onto Purkinje cells in the cerebellar flocculus (*Nguyen-Vu et al., 2013*; *Kimpo et al., 2014*). Adeno-associated virus (AAV) carrying ChR2 under the CaMKIIα promoter (CaMKIIα-ChR2(H134R)-EYFP (*Gradinaru et al., 2007*) was obtained from the Neuroscience Gene Vector and Virus Core at the Stanford School of Medicine. Mice were anesthetized as described above and a craniotomy was made above the inferior olive using stereotaxic coordinates. One-half to 1 ul of the AAV solution was injected into the dorsal cap of Kooy over the course of 15–30 min. Following injection, a minimum of four weeks was allowed for viral expression before further surgeries. Mice then underwent surgical implantation of an eye coil and headpost as described above, during which a craniotomy was also made in the periotic capsule, and a cannula (Plastic One, Inc.) implanted, to allow fiber optic access to the cerebellar flocculus.

## Lentivirus vectors

Lentivirus was used to drive expression of H2-D$^b$ in cerebellar Purkinje cells. H2-D$^b$ (*Garstka et al., 2007*) was cloned into the BamHI site in the fourth exon of the L7/pcp-2 gene (*Zhang et al., 2001*). L7::H2-D$^b$ or L7::GFP (Oberdick lab) was cloned into a pCDH-EF1-MCS-T2A-copGFP lentiviral vector backbone (System Biosciences, Mountain View, CA; where EF1 is the promoter for elongation factor 1α, MCS is the multiple cloning site, T2A is a self-cleaving peptide that allows expression of multiple proteins from a single transcript, and copGFP is a green fluorescent protein used as a reporter),

using SwaI restriction sites, with the EF1 promoter removed to generate the following constructs: L7::H2-D$^b$-T2A-copGFP, L7::H2-D$^b$-(stop)-T2A-copGFP, and L7::GFP-T2A-copGFP.

In the L7::H2-D$^b$-T2A-copGFP construct, the T2A enables separate expression of H2-D$^b$ and copGFP proteins, and is indicated as L7:: H2-D$^b$/GFP. Because a few amino acid residues from the T2A would be left on H2-D$^b$ expressed from the L7::H2-D$^b$-T2A-copGFP virus, a second construct was designed, L7::H2-D$^b$-(stop)-T2A-copGFP, which used the stop codon to allow H2-D$^b$ expression without any additional amino acid residues from the T2A, and without expression of copGFP.

There was no significant behavioral difference between DKO mice injected with L7::H2-D$^b$-T2A-copGFP (n = 7) and L7::H2-D$^b$-(stop)-T2A-copGFP (n = 8) (VOR-increase learning at 1.0 Hz: p=0.23); both viruses rescued the learning deficit in the DKO mice, therefore the data from these two groups were pooled and indicated as L7::H2-D$^b$. The L7::GFP-T2A-copGFP expressed GFP and copGFP, as a negative control for the L7 promoter, and is indicated as L7::GFP.

All three L7 constructs were sequenced and packaged by the Neuroscience Gene Vector and Virus Core at the Stanford School of Medicine. Lentiviruses were produced by polyethyleneimine-mediated transfection of 293 T cells with four separate plasmids encoding HIV-1 gag-pol, HIV-1 rev, VSV-G envelope, and the HIV-1 based genome vector. Virus-containing culture media was harvested 24 and 48 hr post-transfection, filtered through a 0.45 µm filter and concentrated by ultracentrifugation. Concentrated virus was stored in single-use aliquots at −80°C.

A fourth virus, CMV::H2-D$^b$-HA, provided a positive control for detection of H2-D$^b$ in the cerebellum in initial validation experiments, but because its expression was not restricted to Purkinje cells, it was not used further in this study. An HA tag was fused to the c-terminus of H2-D$^b$ by PCR. The CMV promoter was PCRed from a pcDNA3 vector (Thermo Fisher Scientific). CMV::H2-D$^b$-HA was then cloned into a pCDH lentivector backbone (System Biosciences). The construct was sequenced for verification and packaged into lentivirus by the Neuroscience Gene Vector and Virus Core at the Stanford School of Medicine.

## Reverse-transcriptase (RT) PCR

Two to three weeks after virus injection, the cerebellar flocculi of two DKO mice injected with L7::H2-D$^b$-T2A-copGFP virus were dissected for mRNA analysis by RT-PCR. Thalamus from one WT control and spleen from one DKO mouse were used as positive and negative control samples, respectively. Primers for H2-D$^b$ were designed to detect exon 2 and exon 3 regions of H2-D$^b$. RNA was extracted from each sample using RNAqueous-4PCR (Ambion, Life Technologies, NY) and cDNA was synthesized using the iScript cDNA Synthesis Kit (Bio-Rad). PCR products were evaluated by gel electrophoresis to confirm the presence of PCR products of predicted size of ~250 bp. H2-D$^b$ primers:

Sense- 5'CAAGAGCAGTGGTTCCGAGTGAG-3';
Antisense- 5'CTTGTAATGCTCTGCAGCACCACT-3'.

Reactions for RT-PCR were carried out as previously described (*Lee et al., 2014*) using 1 ug of cDNA as a template (5 min at 95°C followed by 40 cycles (30 s at 95°C, 30 s, at 60°C, 30 s, 72°C)) (Veriti 96-well Thermal Cycler, Applied Biosystems). Glyceraldehyde-3-phosphate dehydrogenase (GAPDH) was used as internal control. GAPDH primers: Sense- 5'ATTGTCAGCAATGCATCCTGC-3' Antisense- 5'AGACAACCTGGTCCTCAGTGT-3'. The quality of cDNAs was confirmed by genotyping PCR reactions as described previously (*Lee et al., 2014*) using 0.5 ~ 1 µg of cDNA as template and the samples containing genomic DNAs were discarded.

## Immunohistochemistry

Two weeks after virus injection, three DKO mice injected in the flocculus with L7::H2-Db-T2A-copGFP were deeply anesthetized and immediately perfused with 0.1M PBS followed by 4% paraformaldehye (w/v) in PBS. The brain was removed and post-fixed for 2 hr at 20°C. After fixation, the brain was placed in 30% sucrose (w/v) in PBS solution overnight at 4°C. Then brain was then embedded in OTC (Sakura Fine Tek) and frozen for cryosectioning. Coronal sections of 20 um were made through the cerebellar flocculi. Slices were then incubated for 1 hr at 20°C with blocking solution containing 10% normal goat serum and 1% BSA in PBS with 0.3% Triton-X (PBST), and then overnight at 4°C with primary antibodies diluted in blocking solution (anti-TurboGFP, Evrogen AB513). Slices were then washed three times with PBST and incubated for 1 hr at 20°C with secondary

antibody (Alexa Fluor 488 goat anti rabbit, Invitrogen A2153). Fluorescence images were taken using a Nikon Eclipse E800 microscope.

## Behavioral experiments

During each behavioral experiment, the head of the mouse was immobilized by attaching the implanted head post to a restrainer. The restrainer was attached to a turntable (Carco Electronics, Pittsburgh, PA), which delivered a vestibular stimulus by rotating the mouse about an earth-vertical axis. Visual stimuli were delivered by moving an optokinetic drum made of white translucent plastic with black and white vertical stripes, each of which subtended 7.5° of visual angle. The optokinetic drum was back-lit by fiber-optic lights.

Horizontal eye position was measured using the eye coil method, and sampled at a rate of either 500 or 1000 Hz. Eye velocity was calculated by differentiating eye position measurements obtained from the eye coil. Any data segment containing a saccade or motion artifact was excluded from the analysis and then a sinusoid was fit to remaining data to extract the amplitude and phase of the eye velocity response to vestibular or visual stimuli. The VOR was measured as the eye-movement response to a sinusoidal vestibular stimulus (rotation of the head about an earth-vertical axis, 1.0 Hz or 0.6 Hz, ±10°/s peak velocity) in complete darkness (i.e., with no visual inputs), in 40 s blocks. The VOR gain was calculated as the ratio of the eye-to-head movement amplitudes. The optokinetic reflex (OKR) was measured as the eye-movement response to an optokinetic visual stimulus (1.0 Hz, ±10°/s peak velocity) delivered with the head stationary, and the gain of the OKR was calculated as the ratio of eye-to-visual stimulus amplitudes.

Training to increase the VOR gain consisted of pairing a ± 10°/s sinusoidal vestibular stimulus with oppositely directed ±10°/s sinusoidal optokinetic drum rotation, such that a VOR gain of 2 would be required to stabilize the image on the retina (*Figure 1D*). Training or pre-training to decrease the VOR gain consisted of pairing a ± 10°/s sinusoidal vestibular stimulus with ±10°/s sinusoidal rotation of an optokinetic drum (visual stimulus) in the same direction as the head, such that the visual stimulus was stationary relative to the mouse and required a VOR gain of zero to stabilize the image on the retina (*Figure 1E*). Vestibular-only pre-training consisted of 1 Hz, ±10°/s sinusoidal vestibular stimulation in the dark (i.e., no visual stimulus). Training and pre-training were each conducted in three ten-minute blocks. Before and after each block, the eye movement response to the vestibular stimulus alone, i.e., the VOR, was measured in total darkness, and learning was calculated as the change in VOR gain after each block of training. The frequency of the visual-vestibular training stimuli and the vestibular testing stimuli were either 1.0 Hz or 0.6 Hz.

Experimenters running the behavioral experiments were blind to the genotype of the mice. Individual mice were used for multiple behavioral experiments (subjected to different training and testing conditions), separated by at least two days, with no specific randomization in the order of experiments. If the longevity of the eye coil and head post implants allowed, some mice underwent the same type of behavioral experiment more than once, in which case, the results from the replications were averaged for that animal and a single averaged value was used in the group analysis.

## Optical activation of climbing fibers

In some experiments, animals were subjected to 30 min of optogenetic climbing fiber stimulation prior to visual-vestibular VOR-increase or VOR-decrease training. Experiments were performed on WT mice injected in the inferior olive with virus carrying ChR2, and sham control mice that were not injected with any virus, but underwent that same surgical procedure of eye coil, head post and cannula implantation for behavioral testing. In both the experimental and sham control groups, optical stimulation was delivered unilaterally to the cerebellar flocculus through the implanted cannula via a 200 um optical fiber connected to a blue laser (473 nm, Laserglow). The optical stimulation consisted of 250 ms trains of 3 pulses, repeated every 1 s, with each pulse 2 ms in duration at an intensity of ≤3 mW. The optical stimulation was delivered in 10 min blocks over the course of 30 min while the animal was head-restrained and stationary in the dark. The gain of the VOR was assessed before and after each block of optical stimulation. VOR training began within 2 min of the end of the 30 min of optical stimulation, using vestibular-visual stimulus pairing to either increase or decrease the gain of the VOR.

## Statistical analysis

Adequate sample size was determined based on previous experiments of VOR behavior (*Boyden and Raymond, 2003*; *Boyden et al., 2006*) and optogenetic stimulation (*Nguyen-Vu et al., 2013*) in mice, as borne out the by results. Statistical analyses were performed using Microsoft Excel and Graphpad Prism (RRID:SCR_002798). Data are presented as means ± s.e.m unless otherwise indicated. The Kolmogorov-Smirnov test was used to test for normality, and Bartlett's multiple sample test was used to determine equal variance. Unpaired Student's *t* tests (2-sided) were used to compare groups. When the time course data were compared, two-factor repeated measure ANOVA was used to test for a significant difference between groups. A Fischer's LSD post-hoc test was used only when there was a significant difference between groups. For all tests, p<0.05 was considered to be statistically significant.

## Computational models

We used a computational approach to determine the essential features of synaptic models that could account qualitatively for our central empirical observations: (1) without pre-training, enhanced plasticity impairs learning; (2) pre-training rescues learning in mice with enhanced plasticity; (3) pre-training impairs learning in WT mice; and (4) with appropriate pre-training, mice with enhanced plasticity learn faster than WT (*Figure 4*, *left, reproduced from Figure 3A*). In all cases, the contribution of pf-Pk LTD to VOR learning was modeled by measuring the initial rate of VOR-increase learning with the rate at which synapses transitioned to a depressed state during training(*Ito, 1972*). This rate was determined by three factors: (1) $f^{dep}$, the rate of candidate LTD events, or the pattern of neural activity with the potential to induce LTD (i.e., near simultaneous activation of cerebellar parallel fibers and climbing fibers); (2) $q^{dep}$, the intrinsic plasticity rate, which corresponded to the threshold for LTD induction, or the probability that a candidate LTD event would cause an eligible synapse to transition to a depressed state; and (3) $p_{strong}$, the number of synapses eligible to transition to a depressed state.

The lower induction threshold in DKO mice was modeled as an increase in the intrinsic plasticity rate, $q^{dep(MHC)} > q^{dep}$. Training to increase the VOR was modeled as an increase in the rate of candidate LTD events, $f^{dep} \rightarrow f^{dep} + \Delta f$. Instead of explicitly modeling VOR-decrease learning, because its mechanism(s) are still unknown, we only modeled the component that reverses VOR-increase as an increase in the rate of LTP events: $f^{pot} \rightarrow f^{pot} + \Delta f$. For illustration, we chose $f^{pot} = f^{dep} = \frac{1}{2}$ in the absence of VOR training, though none of our results depend upon this balanced rate of candidate LTP and LTD events. We compared synaptic models with different numbers of potentiated and depressed states and different probabilities of transitioning between states. See Appendix for details.

## Code availability

Computational models were simulated using custom MATLAB (RRID:SCR_001622) code (MATLAB R2013b, The MathWorks Inc., Natick, Massachusetts). The code for simulating the computational models is publicly available.

## Acknowledgements

Data reported in the paper can be made available upon request to the corresponding author. The code for simulating the computational models is publicly available at https://github.com/ganguli-lab/Saturation_Enhanced_Plasticity. We thank J Rinaldi for help on climbing fiber stimulation; SL Shin for advice on the VOR reversal experiment; S Umamoto and R Hemmati for technical support; A Katoh for discussions, assistance and advice; J Oberdick for the L7/pcp2 promoter and L7::GFP plasmids; S Sebastian for the H2-Db construct; and all members of the JLR laboratory for their support. Funding: This study was supported by NIH RO1DC04154, RO1NS072406, R21NS057488 and P30DC10363 and the James S McDonnell Foundation (JLR), an NSF Graduate Research Fellowship and NIH F31DC010547 (TBN), NIH F32NS058060 (GZ), NIH RO1MH07166 (CJS), the Genentech Foundation (SL), the Burroughs Wellcome Foundation (SG) and NIH NS069375 (Stanford Neuroscience Virus Core).

# Additional information

## Competing interests

JLR: Reviewing editor, *eLife*. The other authors declare that no competing interests exist.

## Funding

| Funder | Grant reference number | Author |
|---|---|---|
| National Science Foundation | Graduate Research Fellowship | TD Barbara Nguyen-Vu |
| National Institutes of Health | F31DC010547 | TD Barbara Nguyen-Vu |
| National Institutes of Health | F32NS058060 | Grace Q Zhao |
| Genentech Foundation | | Hanmi Lee |
| Burroughs Wellcome Fund | | Surya Ganguli |
| National Institutes of Health | RO1MH07166 | Carla J Shatz |
| National Institutes of Health | RO1DC04154 | Jennifer L Raymond |
| James S. McDonnell Foundation | | Jennifer L Raymond |
| National Institutes of Health | NS069375 | Jennifer L Raymond |
| National Institutes of Health | RO1NS072406 | Jennifer L Raymond |
| National Institutes of Health | R21NS057488 | Jennifer L Raymond |
| National Institutes of Health | P30DC10363 | Jennifer L Raymond |

The funders had no role in study design, data collection and interpretation, or the decision to submit the work for publication.

## Author contributions

TDBN-V, Conceptualization, Data curation, Formal analysis, Funding acquisition, Investigation, Methodology, Writing—original draft, Project administration, Writing—review and editing; GQZ, Conceptualization, Data curation, Formal analysis, Investigation, Methodology, Writing—original draft; SL, Conceptualization, Software, Formal analysis, Funding acquisition, Investigation, Methodology, Writing—original draft, Writing—review and editing; RRK, Data curation, Writing—review and editing; HL, Data curation, Methodology, Writing—review and editing; SG, Resources, Software, Formal analysis, Supervision, Writing—original draft, Writing—review and editing; CJS, Resources, Supervision, Funding acquisition, Writing—review and editing; JLR, Conceptualization, Resources, Supervision, Funding acquisition, Writing—original draft, Writing—review and editing

## Author ORCIDs

TD Barbara Nguyen-Vu, http://orcid.org/0000-0002-4708-1982
Subhaneil Lahiri, http://orcid.org/0000-0003-2028-6635
Jennifer L Raymond, http://orcid.org/0000-0002-8145-747X

## Ethics

Animal experimentation: All experimental procedures were approved by the Administrative Panel on Laboratory Animal Care at Stanford University under animal care and use committee (IACUC) Protocol #9143, titled 'Vestibular and Visual Control of Eye Movements in Mice'.

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

## Appendix 1

# Mathematical formalism

## Models of synapses

We make the following assumptions:

- The synapses are identical and have $M$ internal functional states, where the states of different synapses are independent of each other.
- Candidate potentiating/depressing plasticity event timings are distributed as Poisson processes with rates $r^{\text{pot/dep}}$.
- Candidate potentiation and depression events are described by Markov processes with transition probabilities $\mathbf{M}^{\text{pot/dep}}$.
- The synaptic weights of the internal states are given by the column vector $\mathbf{w}$. This can only take values in a finite range that we can shift to $\pm 1$. Most of the models will only use the two extreme values.

The overall rate of candidate plasticity events will only affect the units we measure time in. For the questions we are investigating, qualitative understanding of when learning is enhanced/impaired rather than detailed numerical matching of learning curves, this is completely irrelevant. Only the relative rates will matter. With this in mind, we define

$$r = r^{\text{pot}} + r^{\text{dep}}, \quad f^{\text{pot}} = \frac{r^{\text{pot}}}{r}, \quad f^{\text{dep}} = \frac{r^{\text{dep}}}{r},$$

where $f^{\text{pot/dep}}$ is the fraction of candidate plasticity events that are potentiating/depressing.

The independence and identicalness of synapses means that the state of the system can be completely described by the probability distribution over the internal states, the *row* vector $\mathbf{p}(t)$ which evolves as

$$\frac{\mathbf{dp}(t)}{\mathbf{d}t} = r\mathbf{p}(t)\mathbf{W^F}, \quad \mathbf{W^F} = f^{\text{pot}}\mathbf{M^{pot}} + f^{\text{dep}}\mathbf{M^{dep}} - \mathbf{I}, \tag{1}$$

where $\mathbf{W^F}$ is a continuous time Markov transition matrix and $\mathbf{I}$ is the identity matrix. Eventually, this will settle into the equilibrium distribution:

$$\mathbf{p}^{\infty}\mathbf{W^F} = 0. \tag{2}$$

We model the DKO mice by changing $\mathbf{M}^{\text{dep}}_{\text{WT}}$ to $\mathbf{M}^{\text{dep}}_{\text{DKO}}$, which has larger off-diagonal matrix elements. This makes it easier to induce a decrease in synaptic weight, modelling the observed lower threshold for LTD (*McConnell et al., 2009*).

We will look at several different models, the serial model (see *Leibold and Kempter, 2008*; *Ben-Dayan Rubin and Fusi, 2007* and *Appendix 1—figure 1A*) which has only two values for the synaptic weight, the two-state model (which can be thought of as a special case of the serial model, see *Appendix 1—figure 1B*), the multistate model (see *Amit and Fusi, 1994*; *Fusi and Abbott, 2007a* and *Appendix 1—figure 1C*) which has a linearly varying synaptic weight, and the cascade model (see *Fusi et al., 2005* and *Appendix 1—figure 1D*). We will also look at a new, pooled resource model and a non-uniform version of the multistate model that we will define below.

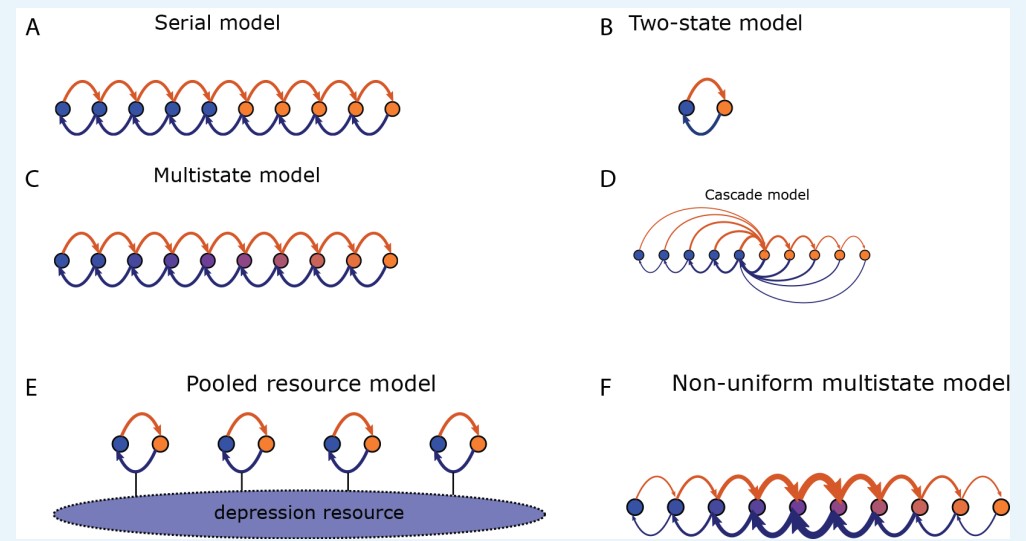

**Appendix 1—figure 1.** Transition probabilities for different models. Potentiation induces transitions indicated by orange arrows, depression indicated by blue arrows. States of strong/weak synaptic weight indicated by orange/blue circles. (**A**) In the serial model the transition probabilities for potentiation/depression are all equal and it is parameterised by these two values. The synaptic weight takes only two values, $\pm 1$. (**B**) The two-state model is parameterised by the two transition probabilities. (**C**) In the multistate model the transition probabilities for potentiation/depression are all equal and it is parameterised by these two values. The synaptic weight varies linearly in the interval $[-1, 1]$. (**D**) In the cascade model, the transition probabilities decay geometrically with a parameter $x$ (see *Fusi et al., 2005*) and synaptic weight takes only two values. (**E**) In the pooled resource model, Several two-state synapses share resources that are required for potentiation and depression. These resources are depleted as more synapses are potentiated or depressed. This pool of synapses can be modelled as one compound synapse. (**F**) In the non-uniform multistate model the synaptic weight varies linearly in the interval $[-1, 1]$, similar to the multistate model, but the transition probabilities between adjacent states decays exponentially away from the central transition for both potentiation and depression.

For the serial, multistate and two state models, we will use the same value for $q$, the transition rate between adjacent states, for potentiation in the wild-type and DKO models. We will use a larger value for $q$ for depression in the DKO models than in the wild-type.

For the cascade model, we will use the same value for the parameter $x$ (which controls the ratio of adjacent transition rates, see *Fusi et al., 2005*) for potentiation in the wild-type and DKO models. We will use a larger value for $x$ for depression in the DKO models than in the wild-type.

The values used for all these parameters in simulations are listed in *Appendix 1—table 1*.

**Appendix 1—table 1.** Parameters used for simulations. For the serial, two state and multistate models, the plasticity parameter listed is the transition probability between adjacent states. For the pooled resource model, the plasticity parameters are the minimum and maximum transition probability for the constituent two-state synapses. For the cascade and non-uniform multistate models, the plasticity parameter is the ratio of adjacent transition probabilities.

| Model | # states | Plasticity parameter | | | $f^{\mathrm{dep}}$ | | | |
| | | Pot | WT dep | DKO dep | Base | Inc | Dec | $rt_{\mathrm{pre}}$ |
| --- | --- | --- | --- | --- | --- | --- | --- | --- |
| Serial | 10 | 0.12 | 0.14 | 0.2 | 0.5 | 0.89 | 0.11 | 100 |

*Appendix 1—table 1 continued on next page*

*Appendix 1—table 1 continued*

| Model | # states | Plasticity parameter | | | $f^{\mathrm{dep}}$ | | | |
| --- | --- | --- | --- | --- | --- | --- | --- | --- |
| | | Pot | WT dep | DKO dep | Base | Inc | Dec | $rt_{\mathrm{pre}}$ |
| Two-state | 2 | 0.1 | 0.1 | 0.2 | 0.5 | 0.6 | 0.4 | 5 |
| Multistate | 10 | 0.3 | 0.3 | 0.4 | 0.5 | 0.8 | 0.2 | 5 |
| Pooled res. | 7 | 0.008 | [0.0006, 0.6] | [0.001, 1] | 0.5 | 0.9 | 0.1 | 20 |
| Cascade | 14 | 0.386 | 0.398 | 0.466 | 0.478 | 0.63 | 0.002 | 200 |
| Non-uni. | 12 | 0.4 | 0.4 | 0.53 | 0.5 | 0.7 | 0.1 | 500 |

## Pooled resource model

Suppose that there is some resource required for potentiation/depression that is shared between $P$ synapses and is depleted as more synapses are potentiated/depressed and replenished when this is reversed, as shown in *Appendix 1—figure 1E*. We can avoid going beyond the independent synapse model by modelling the pool of synapses that share the resource as a single compound synapse.

We will model the individual synapses with the two-state model. Let $i = 0 \ldots P$ be the number of them that are potentiated. We will model the effect of resource depletion linearly with the potentiation/depression probabilities for the individual synapses:

$$
\begin{aligned}
q^{\mathrm{pot}} &= \frac{(P-i-1)q_{\max}+iq_{\min}}{P-1}, & i &= 0 \ldots P-1, \\
q^{\mathrm{dep}} &= \frac{(i-1)q_{\max}+(P-i)q_{\min}}{P-1}, & i &= 1 \ldots P.
\end{aligned}
$$

At each plasticity event for the compound synapse, one of the individual synapses will be chosen randomly for update. This effectively reduces the transition probabilities by $1/P$.

This compound synapse would seem to have $2^P$ internal states. However, we need only keep track of the number of potentiated synapses, not their identity. We can group those states of the pool of synapses that have the same total number of potentiated synapses into a single state of the equivalent compound synapse, leaving $M = P + 1$ states. The transition network will then have a multistate topology (see *Appendix 1—figure 1C*) but the transition probabilities will no longer be uniform and the weight of the compound synapse is the mean of its constituent synapses:

$$
\mathbf{w}_i = \frac{2i}{P} - 1.
$$

The Markov process is lumpable when partitioning states by total number of potentiated synapses (see *Kemeny et al., 1960*; *Burke and Rosenblatt, 1958*; *Ball and Yeo, 1993*). The transition probabilities between lumps $i$ and $j$ is computed by choosing one state from lump $i$ and summing the transition probabilities to all states in lump $j$, which must be the same for all states in lump $i$.

For any state in lump $i$, there are $P - i$ synapses that can be potentiated to go to lump $i + 1$. Each of these transition probabilities is $q^{\mathrm{pot}}/P$. Similarly, there are $i$ synapses that can be depressed to go to lump $i - 1$. Each of these transition probabilities is $q^{\mathrm{dep}}/P$. Thus:

$$
\begin{aligned}
\mathbf{M}^{\mathrm{pot}}_{i\,i+1} &= \left[\frac{(P-i-1)q_{\max}+iq_{\min}}{P-1}\right]\frac{P-i}{P}, & i &= 0 \ldots P-1, \\
\mathbf{M}^{\mathrm{dep}}_{i\,i-1} &= \left[\frac{(i-1)q_{\max}+(P-i)q_{\min}}{P-1}\right]\frac{i}{P}, & i &= 1 \ldots P,
\end{aligned}
$$

with all other off-diagonal elements equal to zero. The diagonal elements are chosen so that the rows sum to one.

This model is parameterised by the range of values, $q \in [q_{\min}, q_{\max}]$, for potentiation and depression. We will use the same values for potentiation in the wild-type asnd DKO models. We will use larger values for depression in the DKO models than in the wild-type. We consider a version of this model that has resource depletion for depression only, so that potentiation transition probabilities are unaffected by the number of potentiated synapses. This is done by setting $q_{\max}^{\mathrm{pot}} = q_{\min}^{\mathrm{pot}}$. The values of these parameters are listed in *Appendix 1—table 1*.

## Non-uniform multistate model

This model, shown in *Appendix 1—figure 1F*, is similar to the multistate model (see *Appendix 1—figure 1C*), as it only has transitions between adjacent states and a linearly varying synaptic weight. However, like the cascade model (see *Fusi et al., 2005* and *Appendix 1—figure 1D*), the transition probabilities decay exponentially away from the central transition. More precisely:

$$\mathbf{M}_{ii+1}^{\mathrm{pot}} = \mathbf{M}_{i+1i}^{\mathrm{dep}} = x^{|\frac{M+1}{2} - i| + \frac{1}{2}}, \qquad i = 1 \ldots M - 1,$$

with all other off-diagonal elements equal to zero. The diagonal elements are chosen so that the rows sum to one.

This model is parameterised by the values of $x$ chosen for potentiation and depression. We will use a larger value for depression in the DKO models. The values used are listed in *Appendix 1—table 1*.

## Model of VOR learning experiment

Training the animal will not change the internal dynamics of a synapse under potentiation or depression. It will change the environment, which will lead to a change in how often potentiation and depression occur. We will model this by changing $f^{\mathrm{pot/dep}}$, leaving $\mathbf{M}^{\mathrm{pot/dep}}$ unchanged. This approach was used to model motor learning by *Smith et al., 2006*.

The untrained case will be described by $f^{\mathrm{dep}} = f_0^{\mathrm{dep}}$. Gain-increase training will be described by $f^{\mathrm{dep}} = f_{\mathrm{inc}}^{\mathrm{dep}} > f_0^{\mathrm{dep}}$, and gain-decrease training will be described by $f^{\mathrm{dep}} = f_{\mathrm{dec}}^{\mathrm{dep}} < f_0^{\mathrm{dep}}$. Note that the forgetting matrix *Equation (1)* and the equilibrium distribution *Equation (2)* depend on $f^{\mathrm{dep}}$, which we will indicate with subscripts.

Before training, the synaptic distribution will be in the equilibrium distribution corresponding to $f_0^{\mathrm{dep}}$. During gain-increase training, it will evolve according to *Equation (1)* with $f_{\mathrm{inc}}^{\mathrm{pot}}$:

$$\mathbf{p}(t) = \mathbf{p}_0^\infty \exp\left(rt\mathbf{W}_{\mathrm{inc}}^{\mathrm{F}}\right).$$

On the other hand, if the gain-increase training follows gain-decrease pre-training for some time period, $t_{\mathrm{pre}}$:

$$\mathbf{p}(t) = \mathbf{p}_0^\infty \exp\left(rt_{\mathrm{pre}}\mathbf{W}_{\mathrm{dec}}^{\mathrm{F}}\right) \exp\left(r(t - t_{\mathrm{pre}})\mathbf{W}_{\mathrm{inc}}^{\mathrm{F}}\right).$$

We will describe the effect of training by the decrease in mean synaptic weight:

$$L(t) = (\mathbf{p}(0) - \mathbf{p}(t))\mathbf{w}. \tag{3}$$

One can approximate the input to a Purkinje cell as some linear combination of the synaptic weights (weighted by the activities of the corresponding parallel fibres). If we are

not keeping track of synaptic identity, the most natural linear combination to use would be an equal sum of them all. The behavioural output (VOR gain) will be some unknown, non-linear function of the synaptic weights, so the best we can hope for is to reproduce qualitative features of the experiment, such as whether learning is enhanced or impaired by the mutation or pre-training.

As the knockout produces no change in baseline performance, there must be a compensatory mechanism somewhere else. We will model this compensation as a simple linear offset, as could be produced by another population of neurons/synapses whose effect cancels with these neurons/synapses.

We will assume $f_{\mathrm{WT}}^{\mathrm{dep}} = f_{\mathrm{DKO}}^{\mathrm{dep}}$. This is because the relevant effects of the knockout are well localised to the Purkinje cells, as shown by the rescue data, so the activity of the parallel and climbing fibres should not change very much. Therefore the rates of potentiation and depression should not change very much either.

For the most part, we set $f_0^{\mathrm{dep}} = \frac{1}{2}$, $f_{\mathrm{inc}}^{\mathrm{dep}} = f_0^{\mathrm{dep}} + \Delta f$ and $f_{\mathrm{dec}}^{\mathrm{dep}} = f_0^{\mathrm{dep}} - \Delta f$, with $\Delta f > 0$. We use the same values for wild-type and DKO models for the reasons discussed above. We could adjust $r$ to keep $r^{\mathrm{pot}}$ unchanged, if so desired, but this would only change the overall timescale and would not affect any of the qualitative comparisons that we are concerned with here. The values of these parameters are listed in *Appendix 1—table 1*.

We are also assuming that the relation between VOR gain and mean synaptic weight is the same for DKO mice and wild-type, except for a linear offset to compensate for the difference in equilibrium weights mentioned above. This ensures that the qualitative questions mentioned above (enhancement or impairment of learning) will not be affected.

## Simulation and analysis of models

The features of the experiments shown in *Figure 4A* of the main text that we'd like to capture are:

1. Without pre-training, gain-increase learning is significantly faster in the wild-type than in the DKO mice.
2. For the wild-type, gain-increase learning is significantly faster without pre-training than with it.
3. For the DKO mice, gain-increase learning is significantly faster with pre-training than without it.
4. After pre-training, gain-increase learning is significantly faster in the DKO mice than in the wild-type.

These questions will not be affected by any output nonlinearity, as long as it is monotonic and fixed. *We will not study gain-decrease learning*, as its mechanisms are not fully understood, and known to be different to gain-increase. We are only modelling the effect of gain-decrease training on *these* synapses. We will also not discuss the curvature of the learning curves, as this can be changed by the nonlinear relation between synaptic weight and VOR gain.

We will try to gain some analytic insight to some of these models by looking at the slope of the learning curve at the start of gain-increase training. This is proportional to the net-flux from the states with strong synaptic weight to the weaker states, measured using the transition rates for gain-increase but the equilibrium distribution for either untrained or gain-decrease, assuming that pre-training lasts long enough to reach the equilibrium distribution for gain-decrease.

The learning curves resulting from these simulations can be found in *Appendix 1—figure 2*. The values for all parameters we will use can be found in *Appendix 1—table 1*.

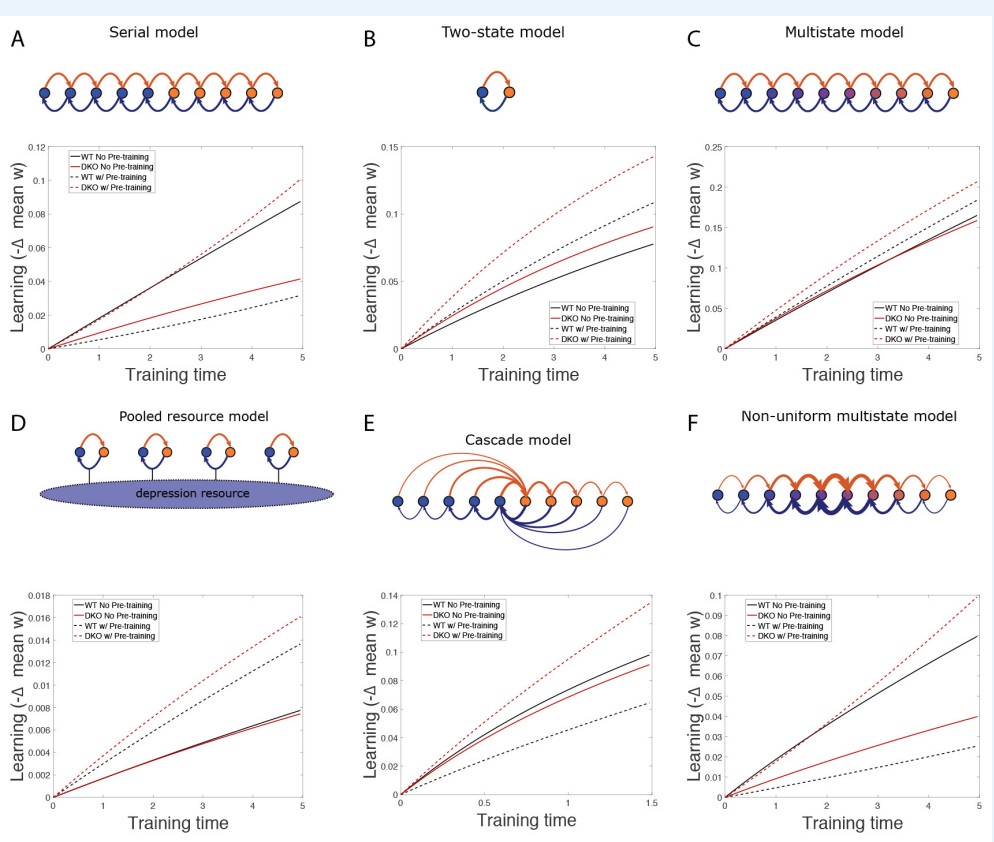

**Appendix 1—figure 2.** Simulation results. (**A–F**) Simulation results for various models, showing decrease in mean synaptic weight over time during gain increase training from the normal state (solid) and after gain-decrease pre-training (dashed) for wild-type (WT; black) and DKO (red) models.

## Serial model

The results of simulations of the serial model can be seen in **Appendix 1—figure 2A**. However, we can get some insight into this model analytically.

Consider the general uniform serial model. Then the equilibrium distribution is given by

$$\mathbf{p}_i^\infty = \frac{1-\alpha}{1-\alpha^M}\alpha^{i-1}, \quad \text{where} \quad \alpha = \frac{f^{\mathbf{pot}}q^{\mathbf{pot}}}{f^{\mathbf{dep}}q^{\mathbf{dep}}}. \tag{4}$$

If we take the limit $\alpha \to 1$, this becomes $\frac{1}{M}$.

The net-flux from the $\mathbf{w} = +1$ states to the $\mathbf{w} = -1$ states is:

$$\Phi = \mathbf{p}_{M/2+1}^\infty f'^{\mathbf{dep}}q^{\mathbf{dep}} - \mathbf{p}_{M/2}^\infty f'^{\mathbf{pot}}q^{\mathbf{pot}} = \frac{1-\alpha}{1-\alpha^M}\alpha^{M/2-1}\left(\frac{\alpha}{\alpha_{\mathbf{inc}}} - 1\right)f_{\mathbf{inc}}^{\mathbf{pot}}q^{\mathbf{pot}}, \tag{5}$$

where $\alpha_{\mathrm{inc}}$ is $\alpha$, but with $f^{\mathrm{pot/dep}}$ replaced by $f_{\mathrm{inc}}^{\mathrm{pot/dep}}$.

To see the effects of the DKO, we can see if this increases or decreases with $q^{\mathrm{dep}}$:

$$\frac{\partial \Phi}{\partial q^{\mathrm{dep}}} = \frac{\alpha^{\frac{M}{2}-1}}{(1-\alpha^M)^2}\left[(1-\alpha^M)-\frac{M}{2}(1-\alpha)(1+\alpha^M)\right]\left(\frac{\alpha}{\alpha_{\mathrm{inc}}}-1\right)\frac{f_{\mathrm{inc}}^{\mathrm{pot}}q^{\mathrm{pot}}}{q^{\mathrm{dep}}}.$$

This quantity can have either sign, depending on $\alpha$ and $M$, so $\Phi$ is not a monotonic function of $q^{\mathrm{dep}}$. Therefore, enhancing plasticity in this way can either enhance or impair the initial learning rate, depending on the parameters.

The effect of pre-training is to replace $f^{\mathrm{pot/dep}}$ with $f_{\mathrm{dec}}^{\mathrm{pot/dep}}$ without changing $f_{\mathrm{inc}}^{\mathrm{pot/dep}}$, which increases $f^{\mathrm{pot}}$ and decreases $f^{\mathrm{dep}}$. Therefore, we can see the effects of pre-training on the initial learning rate by seeing if it increases or decreases with $f^{\mathrm{pot}}$ (with $f^{\mathrm{dep}} = 1 - f^{\mathrm{pot}}$):

$$\frac{\partial \Phi}{\partial f^{\mathrm{pot}}} = \frac{\alpha^{\frac{M}{2}-1}}{(1-\alpha^M)^2}\left[\left(\frac{\alpha}{\alpha_{\mathrm{inc}}}-1\right)\frac{M}{2}(1-\alpha)(1+\alpha^M)-\left(\frac{\alpha^2}{\alpha_{\mathrm{inc}}}-1\right)(1-\alpha^M)\right]\frac{f_{\mathrm{inc}}^{\mathrm{pot}}q^{\mathrm{pot}}}{f^{\mathrm{pot}}f^{\mathrm{dep}}}.$$

This quantity can have either sign, depending on $\alpha$ and $M$, so $\Phi$ is not a monotonic function of $f^{\mathrm{pot}}$. Therefore, pre-training can either enhance or impair the initial learning rate, depending on the parameters.

The lack of monotonicity shown above allows this model to reproduce the key qualitative features of the experiments outlined above. This can be illustrated by looking at a special case, where $q^{\mathrm{pot}} = q^{\mathrm{dep}} = q$ for the wild type, $f_0^{\mathrm{dep}} = \frac{1}{2}$, $f_{\mathrm{dec}}^{\mathrm{dep}} = \frac{1}{2} - \Delta f$ and $f_{\mathrm{inc}}^{\mathrm{dep}} = \frac{1}{2} + \Delta f$.

First, consider the wild-type. Without pre-training:

$$\Phi = \frac{2\Delta f q}{M}. \tag{6}$$

With pre-training:

$$\begin{aligned}\Phi &= 16(\Delta f)^2 q\frac{(1+2\Delta f)^{M/2-1}(1-2\Delta f)^{M/2-1}}{(1+2\Delta f)^M-(1-2\Delta f)^M}\\ &= \frac{4\Delta f q}{M}+\mathcal{O}(\Delta f)^2.\end{aligned}$$

So, we see that pre-training will speed up learning when $\Delta f$ is small. On the other hand, if $\Delta f$ is close to $\frac{1}{2}$, pre-training will initially slow down learning a lot due to the factor of $(1-2\Delta f)^{M/2-1}$.

Intuitively, the flux depends on the slope of the distribution at the centre of the chain (with an offset due to the difference between $f^{\mathrm{pot}}$ and $f^{\mathrm{dep}}$). Pre-training has two effects: it produces a slope in the right direction, but it also reduces the distribution at the centre. For small $\Delta f$, the first effect is stronger and learning speeds up. For larger $\Delta f$, the second effect wins and learning slows down.

This impaired learning after pre-training in the wild-type is caused by the excessive potentiation pushing the synapses away from the central transition that generates the learning signal. Essentially, this model has a form of metaplasticity where repeated potentiation makes subsequent depression harder. This allows excessive pre-training to impair learning, despite increasing the number of potentiated synapses.

Now, consider the DKO models, for which we define $\beta = q^{\mathrm{pot}}/q^{\mathrm{dep}} < 1$, and $q^{\mathrm{pot}} = q$. Without pre-training:

$$\Phi = 2\Delta f q\frac{(1-\beta)\beta^{M/2-1}}{1-\beta^M}. \tag{7}$$

This will be smaller than *Equation (6)* if $\beta < \beta^*(M)$, where $\beta^*(M)$ is defined as the value at which they are equal. This function is plotted in *Appendix 1—figure 3A*, where we can see that it approaches 1 rapidly as we increase $M$.

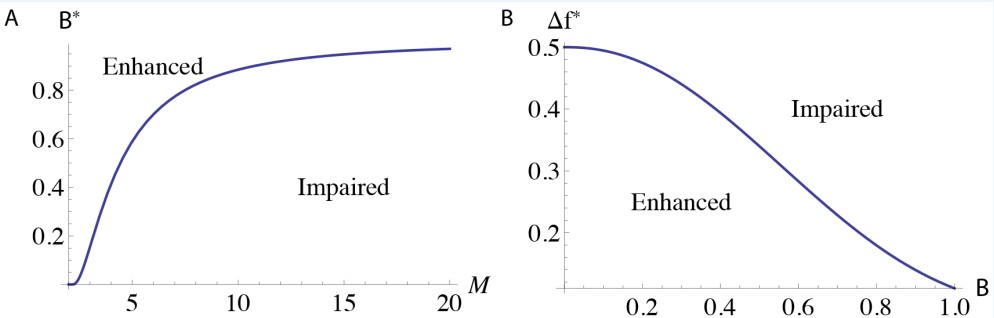

**Appendix 1—figure 3.** Regions of parameter space with different learning outcomes for the serial model. The functions (**A**) $\beta^*(M)$, which describes when the DKO models have impaired learning, and (**B**) $\Delta f^*(\beta, M)$ for $M = 10$, which describes when pre-training enhances learning.

There are two effects here as well. Smaller $\beta$ will increase the probability of crossing the centre of the chain, speeding up learning, but it will also concentrate probability at the ends of the chain, depleting the centre and slowing down learning. The first effect goes like $1/\beta$, whereas the second goes like $\beta^{M/2}$ and will be more significant for smaller $\beta$ or in a longer chain. This exponential decay in the initial distribution is what allows the saturation bias to overwhelm the increased intrinsic plasticity rate and lead to impaired learning with enhanced plasticity. It is important that a learning signal is only generated by the central transition so that the exponential decay can take effect.

With pre-training:

$$\begin{aligned}\Phi &= 4\Delta f q \frac{(1-2\Delta f)-\beta(1+2\Delta f)}{(1-2\Delta f)^M - \beta^M(1+2\Delta f)^M} [\beta(1-2\Delta f)(1+2\Delta f)]^{M/2-1}\\ &= 4\Delta f q \frac{(1-\beta)\beta^{M/2-1}}{1-\beta^M} + \mathcal{O}(\Delta f)^2.\end{aligned}$$

(8)

Once again, we see that pre-training will speed up learning when $\Delta f$ is small, whereas, if $\Delta f$ is close to $\frac{1}{2}$, pre-training will initially slow down learning.

Let us define $\Delta f^*(\beta, M)$ to be the value at which *Equation (7)* and *Equation (8)* are equal. Values of $\Delta f$ that are larger than this (stronger training) will correspond to slower learning after pre-training. As we would like pre-training to slow down learning in the wild-type but speed it up in the DKO models, it would seem that we require $\Delta f^*(1, M) < \Delta f < \Delta f^*(\beta, M)$ (remembering that the wild-type corresponds to $\beta = 1$). As we see in *Appendix 1—figure 3B*, $\Delta f^*(\beta, M) > \Delta f^*(1, M)$, so this range always exists.

In conclusion, if we choose $\beta < \beta^*(M)$, $\Delta f \in [\Delta f^*(1, M), \Delta f^*(\beta, M)]$, we will see that the DKO models learn slower than wild-type without pre-training, and that pre-training speeds up learning in the DKO models but slows it down in the wild-type.

## Two-state model

The results of simulations of the two-state model can be seen in *Appendix 1—figure 2B*. However, this model can be solved exactly:

$$\mathbf{p}^{\infty} = \frac{\left(f^{\mathrm{dep}}q^{\mathrm{dep}} f^{\mathrm{pot}}q^{\mathrm{pot}}\right)}{\lambda}, \quad \mathbf{p}(t) = \mathbf{p}^{\infty} + (\mathbf{p}(0) - \mathbf{p}^{\infty})\mathrm{e}^{-\lambda rt},$$
$$\text{where} \quad \lambda = f^{\mathrm{pot}}q^{\mathrm{pot}} + f^{\mathrm{dep}}q^{\mathrm{dep}}.$$

In practice, it is easier to just substitute $M = 2$ into the formulae in the previous section.

We find that

$$\Phi = \frac{q^{\mathrm{pot}}q^{\mathrm{dep}}\Delta f}{\lambda}, \quad \frac{\partial \Phi}{\partial q^{\mathrm{dep}}} = \frac{f^{\mathrm{pot}}(q^{\mathrm{pot}})^2 \Delta f}{\lambda^2}, \quad \frac{\partial \Phi}{\partial f^{\mathrm{pot}}} = \frac{q^{\mathrm{pot}}q^{\mathrm{dep}}\lambda_{\mathrm{inc}}}{\lambda^2}. \tag{9}$$

The flux is a monotonically increasing function of $q^{\mathrm{dep}}$ and $f^{\mathrm{pot}}$. Therefore, enhancing plasticity in this way, or pre-training, can only enhance the initial learning rate for all values of the parameters, unlike the experimental results shown in ***Figure 4A*** of the main text.

We have only looked at the initial learning rate here, but for the experiments we looked at the learning after a finite period of time. As the total change will be larger for the wild-type than DKO models, it must overtake eventually. If this happens early enough (i.e. between the first and second data point in ***Figure 3B*** of the main text), the initial period where the DKO models learns faster than wild-type would not be seen in the experiment. However, the timescale for this crossover would be similar to the timescale of the exponentials, which is at least as long as the timescale for saturation of learning. In ***Figure 3B*** of the main text we see that this timescale is longer than the gaps between successive measurements, so we would not miss any initial period of enhanced learning in the DKO mice if this were the correct model. In addition, the models without pre-training can never catch up with those with pre-training, and this is sufficient to rule out this model.

This assumed that all synapses have the same parameters. What would happen if there were a population of synapses with different parameters? In this case, the learning rate would be determined by the average flux over this distribution:

$$\bar{\Phi} = \int \Phi\left(q^{\mathrm{dep}}, \ldots\right) p\left(q^{\mathrm{dep}}, \ldots\right) \mathrm{d}q^{\mathrm{dep}} \ldots,$$

where the dependence on all other parameters is omitted.

The DKO mice will have a different distribution, $p_{\mathrm{DKO}}\left(q^{\mathrm{dep}}, \ldots\right)$. It is helpful to introduce a function $P\left(q^{\mathrm{dep}}, \ldots\right)$ that is a cumulative distribution wrt. $q^{\mathrm{dep}}$, but a probability density wrt. all other parameters:

$$P\left(q^{\mathrm{dep}}, \ldots\right) = \int_0^{q^{\mathrm{pot}}} p(q, \ldots) \mathrm{d}q.$$

Then, integration by parts leads to

$$\bar{\Phi} = \langle \Phi(1, \ldots) \rangle - \int \frac{\partial \Phi}{\partial q^{\mathrm{pot}}} P\left(q^{\mathrm{dep}}, \ldots\right) \mathrm{d}q^{\mathrm{dep}} \ldots,$$

where the first term is averaged over all the other omitted parameters and the partial derivative is positive for this model ***Equation (9)***.

The statement that the DKO models have larger $q^{\mathrm{dep}}$ than wild-type is now a statement about the distributions, with the DKO models having more density at larger values. This is not a precise concept, but it will tend to lead to $P\left(q^{\mathrm{dep}}, \ldots\right)$ being smaller for the DKO models than wild type, which makes the second term less negative and the average flux larger, in contradiction with the experimental results in ***Figure 4A*** of the main text.

If $P_{\text{DKO}} \leq P_{\text{WT}}$ at all parameter values and the marginal distribution for all other parameters is the same for both models, then $\bar{\Phi}_{\text{DKO}} \geq \bar{\Phi}_{\text{WT}}$. This is guaranteed if, for example, there is a one-to-one map between $q^{\text{dep}}$ for the two models, so that the cumulative distributions are the same at $q_{\text{WT}}^{\text{dep}}$ and $q_{\text{DKO}}^{\text{dep}} = g\left(q_{\text{WT}}^{\text{dep}}\right)$ for some function $g(x) \geq x$.

If the dependence of the distribution on $q^{\text{dep}}$ is mixed with the other parameters in some complicated way, or if the edges of the distributions are very unusual, it could be possible for the enhanced plasticity models to have an impaired initial learning rate, but this seems contrived and unlikely. A similar argument applies to the effect of pre-training and the distributions over $f^{\text{pot}}$.

The monotonicity shown above means that this model cannot reproduce the key qualitative features of the experiments outlined above, at least as for the initial learning rate.

For this simple binary synapse model, the ratio of the numbers of potentiated and depressed synapses is equal to the ratio of the potentiating and depressing transition rates. This means that the leading effects of enhanced intrinsic plasticity rates and saturation bias cancel each other. This is even true as the number of synapses available for further plasticity approaches zero, which requires that the depressing transition be infinitely stronger than the potentiating transition, compensating for the saturation. There is a requirement that the total number of synapses, potentiated and depressed, is fixed. This normalization effect dampens the effect of saturation bias, so that the effect of enhanced intrinsic plasticity rates dominates for all parameter values. Thus, we must see enhanced initial learning with enhanced plasticity in the binary synapse model.

One can see that pre-training can only increase the fraction of synapses available for depression. This model has no mechanism that could ever result in this causing impaired learning.

Intuitively, this model is missing two features of the serial model. First, it does not have the exponential amplification of the effect of initial saturation bias because this model does not have a chain of states for the distribution to decay across. Second, it does not have the metaplastic effect where repeated potentiation makes future depression harder, as potentiation will merely increase the number of potentiated synapses.

## Multistate model

In this section, we will consider the multistate model as defined in **Amit and Fusi, 1994**, i.e. with linearly varying synaptic weight. The numerical results can be seen in **Appendix 1—figure 2C**, but we can get some analytic insight into this model as well. In essence, this model is like a series of two-state models attached to each other, in contrast to the serial model for which the synaptic weight only changes between one of the pairs of states.

The equilibrium distribution, **Equation (4)**, still applies. However, now the rate of change of our learning metric, **Equation (3)**, will be proportional to the sum of the net fluxes between adjacent states:

$$\Phi = \sum_{i=1}^{M-1} \mathbf{p}_{i+1}^{\infty} f'^{\,\mathbf{dep}} q^{\mathbf{dep}} - \mathbf{p}_i^{\infty} f'^{\,\mathbf{pot}} q^{\mathbf{pot}} = \frac{1-\alpha^{M-1}}{1-\alpha^M}\left(\frac{\alpha}{\alpha_{\text{inc}}}\right) f_{\text{inc}}^{\mathbf{pot}} q^{\mathbf{pot}}. \tag{10}$$

To see the effects of the DKO, we can see if this increases or decreases with $q^{\text{dep}}$:

$$q^{\text{dep}} \frac{\partial \Phi}{\partial q^{\text{dep}}} = \frac{\alpha^{M-1}}{(1-\alpha^M)^2} [1-\alpha] \left[ M - \sum_{i=1}^{M} \alpha^{i-1} \right] \left( \frac{\alpha}{\alpha_{\text{inc}}} - 1 \right) f_{\text{inc}}^{\text{pot}} q^{\text{pot}}. \tag{11}$$

If $\alpha < 1$, both quantities in square brackets are positive, as the sum contains $M$ terms that are all at most 1. If $\alpha > 1$, both of them are negative. Therefore this quantity is positive, so $\Phi$ is a monotonically increasing function of $q^{\text{dep}}$. Therefore, enhancing plasticity in this way can only enhance the initial learning rate, for all values of the parameters.

We can also see the effects of pre-training on the initial learning rate by seeing if it increases or decreases with $f^{\text{pot}}$ (with $f^{\text{dep}} = 1 - f^{\text{pot}}$):

$$f^{\text{pot}} f^{\text{dep}} \frac{\partial \Phi}{\partial f^{\text{pot}}} = \frac{\alpha^{M-2}}{(1-\alpha^M)^2} [1-\alpha] \left( \left[ M - \sum_{i=1}^{M} \alpha^{i-1} \right] + \frac{\alpha}{\alpha_{\text{inc}}} \left[ \sum_{i=1}^{M} \alpha^{1-i} - M \right] \right) f_{\text{inc}}^{\text{pot}} q^{\text{pot}}.$$

This quantity is positive, for similar reasons to *Equation (11)*, so $\Phi$ is a monotonically increasing function of $f^{\text{pot}}$. Therefore, pre-training can can only enhance the initial learning rate, for all values of the parameters.

The monotonicity shown above means that this model cannot reproduce the key qualitative features of the experiments outlined above, at least as for the initial learning rate.

We have only looked at the initial learning rate here, but for the experiments we looked at the learning after a finite period of time. Just as the case of the two state model, the wild-type will eventually catch up with DKO model. This can happen very quickly, as seen in *Appendix 1—figure 2C* (solid curves). This means that the initial period, where the DKO models learns faster than wild-type, might not be seen in the experiment. However, the models without pre-training can never catch up with those with pre-training, and this is sufficient to rule out this model.

Like the two state model, this model is missing two features of the serial model. First, it does not have the amplification of the effect of initial saturation bias as every transition contributes to the learning signal, so the exponentially decaying distribution has no effect. Second, it does not have the metaplastic effect where repeated potentiation makes future depression harder, as potentiation will merely increase the number of potentiated synapses without pushing them away from any boundary between strong and weak states.

## Pooled resource model

The results of simulations of the pooled resource model can be seen in *Appendix 1—figure 2D*. The numerical results can help us understand it qualitatively.

If we compare gain-increase learning in the DKO models to wild-type, there are two effects: the increased transition rates speed up learning, but the equilibrium distribution is shifted to the depressed side, where there are fewer synapses available for depression and resources are depleted. When resource depletion is sufficiently severe, the second effect dominates and the DKO models learn slower than wild-type (see *Appendix 1—figure 2D*, solid curves), which matches what is seen in the experiment.

Gain-decrease pre-training lessens the second effect, and results in the DKO models learning faster than wild-type (see *Appendix 1—figure 2D*, red curves), as seen in the experiment.

Gain-decrease pre-training will shift the distribution to the potentiated side, where there are more synapses available for depression and resources are more plentiful, for both the DKO models and wild type. This means that the pre-trained animals will learn faster then the untrained one for both DKO models and wild-type (see *Appendix 1—figure 2D*). This

differs from what is seen experimentally, where the pre-trained wild-type learns slower than the untrained one.

To verify that this always happens, we scanned over acceptable values of the six parameters relevant to gain-decrease pre-training in wild-type models: $q^{pot}$, $q_{min}^{dep} < q_{max}^{dep}$, and $f_{dec}^{dep} < f_0^{dep} < f_{inc}^{dep}$. Each parameter was scanned over 10 values from 0.05 to 0.95, rejecting values that do not respect the inequalities above. For each parameter set, we computed the initial learning rate, $\dot{L}(0)$ from **Equation (3)**, with and without pre-training. To qualitatively reproduce the experimental results, we would need $\dot{L}_{nopre} - \dot{L}_{pre} > 0$ at $t = 0$. The range of values obtained is plotted in **Appendix 1—figure 4**, where we see that this difference is always negative, ruling out this model.

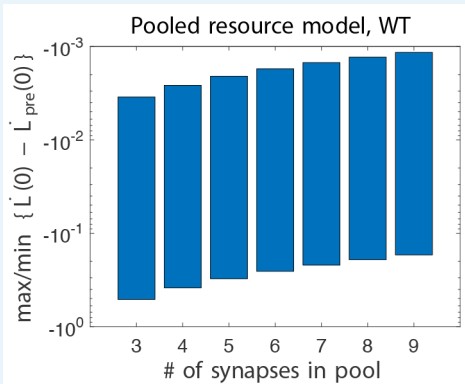

**Appendix 1—figure 4.** Difference in initial learning rate in wild-type pooled resource models, without and with gain-decrease pre-training. Learning, $L(t)$ is defined in **Equation (3)**. Here, we show the maximum and minimum values of this quantity when scanning all six relevant parameters (see text). We see that this quantity is always negative, whereas it was positive in the experiments.

This model shares one feature of the serial model. The effect of initial saturation bias is amplified by resource depletion, allowing for impaired learning with enhanced plasticity. However, it does not have the metaplastic effect where repeated potentiation makes future depression harder, in fact it shows the opposite effect due to resource replenishment, so it can never show impaired learning after pre-training.

## Cascade and non-uniform multistate models

The results of simulations of the cascade and non-uniform multistate models can be seen in **Appendix 1—figure 2E,F**. The numerical results can help us understand them qualitatively.

For the cascade model, we see that the DKO models are slower than wild-type without pre-training but faster with it. This seems to be due to the fact that, without pre-training, very few synapses will be available for depression as most of them are already depressed. With pre-training, some of them will now be potentiated, and the enhanced depression can speed up learning.

Sufficiently long pre-training pushes the synapses far down the cascade, slowing down learning for the wild-type (see **Appendix 1—figure 2E**, black curves). This effect is weaker for the DKO models, as the equilibrium distribution is not as heavily concentrated at the end, as the distribution before pre-training is shifted in the opposite direction.

For the non-uniform multistate model, we see that the DKO models are slower than wild-type without pre-training, as seen experimentally. This seems to be because the enhanced depression results in a greater fraction of synapses starting in states of weaker synaptic

weight where the depressing transitions have lower probability (see *Appendix 1—figure 1F*).

However, gain-decrease pre-training will push synapses towards the states of stronger synaptic weight. The wild-type will be pushed further that the DKO models, as the latter started further to the weaker side as explained above. As the depressing transitions have lower probability for the strongest states (see *Appendix 1—figure 1F*), this, in addition to the enhanced depression of the DKO models, will result in the DKO models learning faster than wild-type, as seen experimentally.

When comparing learning with and without gain-decrease pre-training, there are two effects to consider. First, the synapses are pushing towards the states of stronger synaptic weight, resulting in more synapses being available for depression, speeding up learning. Second, it will also place synapses in states where transitions have lower probability, slowing down learning. The second effect will be weaker in the DKO models than in wild-type, as they started further to the weaker side as explained above. This means that, for appropriate parameter choices, gain-decrease pre-training can speed up learning in the DKO models, but also slow down learning in wild-type, as seen experimentally.

These models share both key features of the serial model. First, the effect of initial saturation bias is amplified by the exponential decay of transition probabilities away from the middle. Second, they both have the metaplastic effect where repeated potentiation makes future depression harder, also due to he exponential decay of transition probabilities.

## Long term effects of pre-training

The fact that after pre-training, gain-increase learning is significantly faster in the DKO models than in the wild-type, as discussed in the previous section, is true of the initial learning rate. With a long enough duration of training, memory of the initial conditions should be erased and the final learning outcome should be the same with or without pre-training. In *Appendix 1—figure 5* we show the result of longer training durations on all models considered above. For all of these models, we see that the wild-type models eventually catch up and overtake the DKO. The initial learning enhancement is a transient effect, and at later times it turns into a learning impairment. This is consistent with an extrapolation of the trend in *Figure 3B* of the main text.

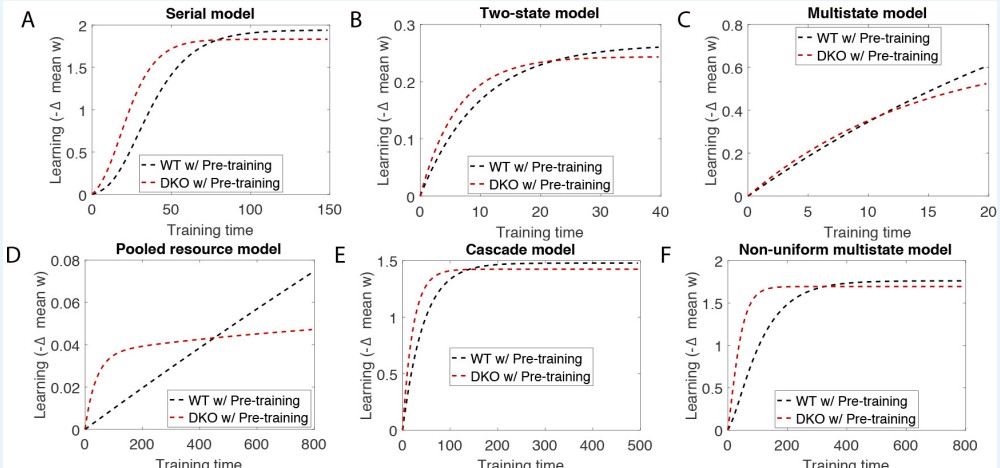

**Appendix 1—figure 5.** Simulation of training for long times after pre-training shows that it erases the effect of pre-training. These simulations were performed with the same parameters as in *Appendix 1—table 1*, but with much longer training times, for the (**A**) serial, (**B**) two-

state, (**C**) multistate, (**D**) pooled resource, (**E** cascade and (**F**) non-uniform multistate models. Compare with *Appendix 1—figure 2*.

## Model of non-specific CF stimulation

Here, we model the effects of non-specific climbing fibre stimulation, as in *Figure 2B* of the main text. It is necessary to also model the 'other' population of synapses that compensate for the change in baseline synaptic weight of the synapses considered up to now, either due to enhanced plasticity or non-specific CF stimulation.

To be concrete, we model these other synapses as described in *Figure 2—figure supplement 2A*. We set the relative rate of depression events for these synapses during training to $f^{\mathrm{dep}} = f_0^{\mathrm{dep}}$, i.e. the baseline value, so that they are unaffected by training. Here we set the rate of plasticity events causing forgetting of the CF stimulation in these synapses is the same as the rate of plasticity events in the original synapses during training, although similar results could be acheived with other choices. For both sets of synapses during non-specific CF stimulation, we set $f^{\mathrm{dep}} > f_0^{\mathrm{dep}}$ to model the increased rate of depression. The values of all parameters used can be seen in *Appendix 1—table 2*.

**Appendix 1—table 2.** Parameters used for simulations. For the serial, two state and multistate models, the plasticity parameter listed is the transition probability between adjacent states. For the pooled resource model, the plasticity parameters are the minimum and maximum transition probability for the constituent two-state synapses. For the cascade and non-uniform multistate models, the plasticity parameter is the ratio of adjacent transition probabilities.

| Model | # states | Plasticity parameter | | $f^{\mathrm{dep}}$ | | | $rt_{\mathrm{pre}}$ |
| | | Pot | Dep | Base | Inc | CF stim | |
|---|---|---|---|---|---|---|---|
| Serial | 10 | 0.12 | 0.14 | 0.5 | 0.89 | 0.9879 | 100 |
| Two-state | 2 | 0.1 | 0.4 | 0.5 | 0.7 | 0.91 | 5 |
| Multistate | 10 | 0.3 | 0.3 | 0.5 | 0.8 | 0.96 | 5 |
| Pooled res. | 7 | 0.08 | [0.006, 0.6] | 0.5 | 0.9 | 0.99 | 20 |
| Cascade | 14 | 0.386 | 0.398 | 0.522 | 0.63 | 0.99 | 200 |
| Non-uni. | 12 | 0.4 | 0.4 | 0.5 | 0.7 | 0.99 | 500 |

In *Appendix 1—figure 6A–F* we see examples of simulations of all of the models we have considered. We see that every single one of them is capable of reproducing the experimental results of *Figure 2B* of the main text, for suitable parameter choices. This is because this intervention has the effect of depleting the population of synapses available for depression *without* the competing effect of enhanced plasticity. Therefore the depletion effect is stronger (than 0) and learning is impaired. This means that these experiments cannot rule out any of the models. However these experiments do agree with the idea of impaired learning due to depletion of the population of labile synapses.

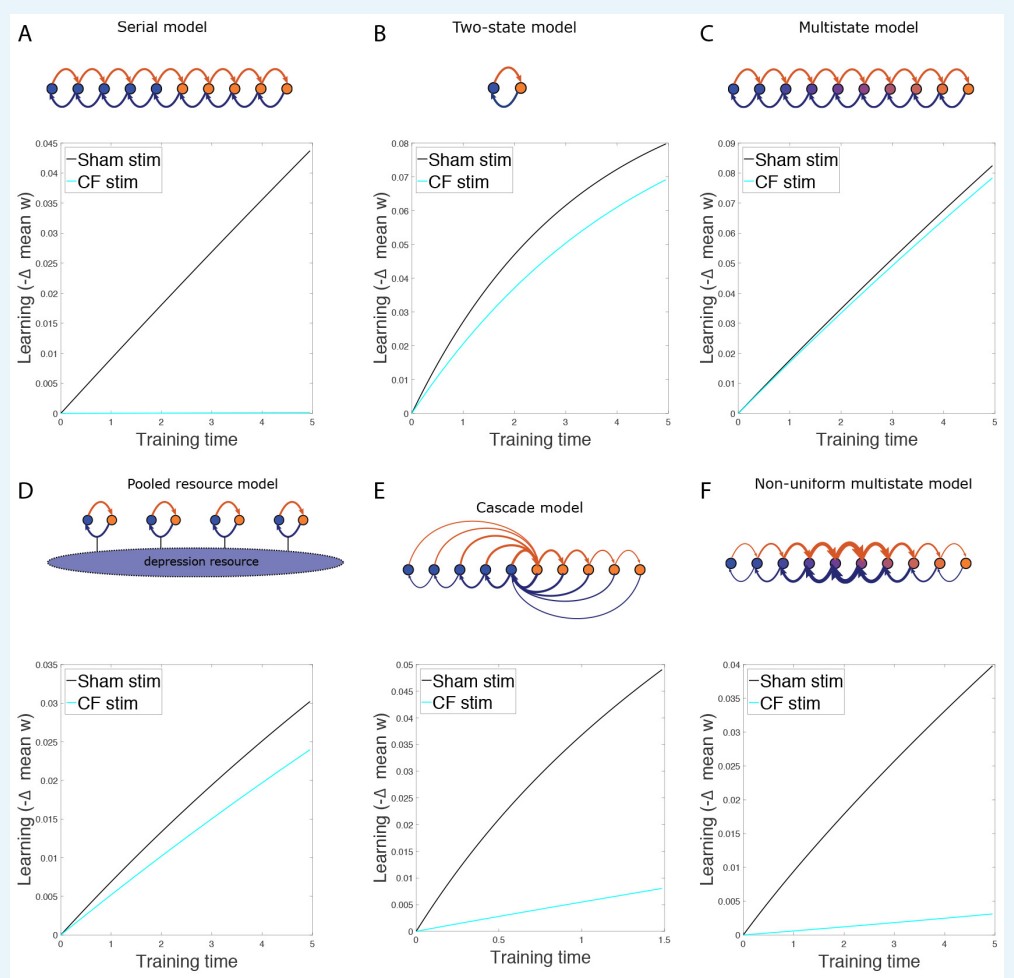

**Appendix 1—figure 6.** The effect of non-specific CF stimulation. (**A**−**F**) Simulation results for various models, showing decrease in mean synaptic weight over time during gain increase training from the normal state (black) and after non-specific CF stimulation (cyan) for wild-type models.

## Balanced potentiation and depression

As the baseline oculomotor performance was normal in the DKO mice (see *Figure 1—figure supplement 2*), one might ask if it is possible that potentiation is also enhanced by the knockout, in such a way that the baseline mean synaptic weight is unchanged.

Intuitively, this seems impossible. The impaired learning observed in some of the models considered depended on the competition between the enhanced intrinsic plasticity rate and the depletion of the population of synapses available for depression. If the second effect is removed, the first must dominate and learning will be enhanced.

However, enhancing potentiation will increase the number of synapses making transitions in the 'wrong' direction, impairing learning. Furthermore, the distribution of synapses across labile and stubborn states may be altered, also affecting the learning rate.

For the serial, two-state and multistate models, the validity of the intuition can be seen analytically. From *Equation (4)*, we can see that the baseline mean synaptic weight depends monotonically on the quantity

$$\alpha = \frac{f^{\mathrm{pot}} q^{\mathrm{pot}}}{f^{\mathrm{dep}} q^{\mathrm{dep}}}.$$

Maintaining the same baseline would then require increasing $q^{\mathrm{pot}}$ in proportion to $q^{\mathrm{dep}}$. As one can see from *Equation (5)*, *Equation (9)* and *Equation (10)*, this will always lead to an enhanced initial learning rate.

For the other models, a numerical analysis is required, as shown in *Appendix 1—figure 7*. To see if learning is impaired with enhanced plasticity each relevant parameter was scanned over 10 values from 0.05 to 0.95 (or from 0.05 to 0.5 for the $x$ parameter of the cascade model), rejecting values that do not respect the inequalities: $0 \le f_0^{\mathrm{dep}} < f_{\mathrm{inc}}^{\mathrm{dep}} \le 1$, for the pooled resource model $0 \le q_{\mathrm{min,WT}}^{\mathrm{dep}} < q_{\mathrm{min,DKO}}^{\mathrm{dep}}, q_{\mathrm{max,WT}}^{\mathrm{dep}} < q_{\mathrm{max,DKO}}^{\mathrm{dep}} \le 1$ and $0 \le q_{\mathrm{WT}}^{\mathrm{pot}} \le 1$, for the cascade model $0 \le x_{\mathrm{WT}}^{\mathrm{dep}} < x_{\mathrm{DKO}}^{\mathrm{dep}} \le \frac{1}{2}$ and $0 \le x_{\mathrm{WT}}^{\mathrm{pot}} \le \frac{1}{2}$, and for the nonuniform multistate model $0 \le x_{\mathrm{WT}}^{\mathrm{dep}} < x_{\mathrm{DKO}}^{\mathrm{dep}} \le 1$ and $0 \le x_{\mathrm{WT}}^{\mathrm{pot}} \le 1$. In each case, the DKO potentiation parameter, $q_{\mathrm{DKO}}^{\mathrm{pot}}$ for the pooled resource model and $x_{\mathrm{DKO}}^{\mathrm{pot}}$ for the others, was adjusted to match the baseline mean synaptic weight. $\mathbf{p}(0)\mathbf{w}$, with wild-type.

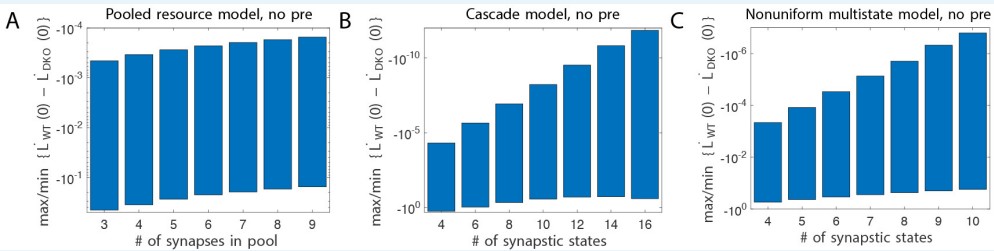

**Appendix 1—figure 7.** Balanced potentiation and depression. Difference in initial learning rate in WT and DKO models, without gain-decrease pre-training. Learning, $\mathrm{L(t)}$ is defined in *Equation (3)*. Here, we show the maximum and minimum values of this quantity when scanning all 7/5 relevant parameters (see text). We see that this quantity is always negative, whereas it was positive in the experiments.

We see that, for all parameter values, such models do not reproduce the observed learning impairment with enhanced plasticity. The normal baseline oculomotor performance must have some other explanation, such as those outlined in *Figure 2—figure supplement 2*.

