## [Decision Letter]

Thank you for submitting your article "Understanding both enhanced and impaired learning with enhanced plasticity: a saturation hypothesis" for consideration by *eLife*. Your article has been reviewed by three peer reviewers, and the evaluation has been overseen by a Reviewing Editor and Timothy Behrens as the Senior Editor. The reviewers have opted to remain anonymous.

The reviewers have discussed the reviews with one another and the Reviewing Editor has drafted this decision to help you prepare a revised submission.

Summary:

While over the years the link between learning and synaptic plasticity has become tighter and tighter, the effect of changes in plasticity can sometimes be puzzling. Nguyen-vu et al. use a relatively simple and well-studied form of cerebellum-dependent motor learning to explore the issue of how genetically enhanced synaptic plasticity impacts learning. Combining behavioral genetics, in-vivo physiology, computational modeling, and mathematical analysis, they conclude that enhanced LTD can lead to both an increase and a decrease of the VOR depending on the history of synaptic changes and the details of the synaptic plasticity model.

Essential revisions:

– The different models used by the authors assume that LTD transition probabilities are increased in KO mice compared to WT, but that LTP transition probabilities are unchanged. As a result, baseline synaptic strengths should be lower in KO than in WT. Some of these authors have presented data in the McConnell et al. 2009 paper, where the amplitude of baseline EPSCs is shown in Figure 3, but it is not fully clear there whether they are identical in WT and KO.

If they are identical, then this data might be consistent with a model in which the LTP transition rates are also increased, so that the baseline synaptic strengths are identical in both types of mice. Does such a model reproduce the data? Alternatively, if the McConnell data show a difference in WT and KO amplitudes, it would be interesting to directly compare this to the model.

– Generally the parameter space of the models is not very well analyzed. So the reader is left with the impression that parameters are chosen such that the effect can or cannot be reproduced. Particularly for the model classes that do not allow analytical arguments a more comprehensive parameter check is necessary, or the choice of the parameters has to justified more convincingly.

– Concerning the optogenetic experiments: If the optogenetic experiments really provide a test of the ideas regarding cascade models, then this experiment should be explicitly modeled using the various different plasticity models discussed. If the optogenetic results are also predicted by standard/simpler models then the discussion of this experiment and its significance should be modified, i.e. toned down, accordingly.

– Are other cell types in cerebellum or vestibular system affected by the mutation? If so this should be mentioned and the implications discussed.

– There are a number of straightforward experimental predictions that naturally follow from the model that are not mentioned by the authors. One straightforward prediction is that with enough VOR-increase training, the advantage of the KO mice should eventually disappear, and in fact the WT should again become better, since the memory of initial conditions should eventually decay away. I am not suggesting the authors perform the corresponding experiments, but I believe it would be good to clarify this issue in simulations.

Related to that, in the Discussion section, the authors write “Within a given training session, VOR learning appears to asymptote within a few tens of minutes”. This does not seem to be true in the experiments reported in Figure 2. These experiments leave open the possibility that these effects are only transient, and that given enough time WT and KO would reach the same level of learning.

Finally, the issue of time scales needs more discussion. For instance, on longer time scales already the 2 state model WT can overtake the KO (the top one in Figure 3B).

– The rates of activity in granule cells in the flocculus in vivo seems important for the modeling but has never been directly measured. Granule cells might be expected to fire very little spontaneously. What rates were assumed in the model? How do conclusions of the model depend on assumed activity levels in granule cells?

– A previous study by McConnell et al. found that rotorod learning and retention was enhanced in the same mutant mice. This seems at odds with the present findings. The authors should discuss possible explanations based on their work.

– The Abstract and Introduction should be rewritten to deal more fairly with the previous literature. While the combination of techniques is novel, many elements are not. The manuscript falls short of acknowledging the fact that many researches since several decades have been studying these types problems from a variety of directions sometimes termed "plasticity stability dilemma". Just to name a few of these approaches:

a) From the theoretical perspective palimpsest models (Amit and Fusi, 1994; and many others) have very early laid out early on that an increase in learning rate also increases forgetting. b) Studies on trafficking of receptor molecules come to similar conclusions (reviewed in Gerrow and Triller, 2010) c) Papers on synaptic metaplasticity and tagging/consolidation raising similar problems

[Editor’s note: Further clarifications were requested on acceptance. The authors’ response follows.]

Congratulations, we are pleased to inform you that your article, "A saturation hypothesis to explain both enhanced and impaired learning with enhanced plasticity", has been accepted for publication in *eLife*.

While over the years the link between learning and synaptic plasticity has become tighter and tighter. The effect of changes in plasticity can sometimes be puzzling. Nguyen-vu et al. use a relatively simple and well-studied form of cerebellum-dependent motor learning to explore the issue of how genetically enhanced synaptic plasticity impacts learning. Combining behavioral genetics, in-vivo physiology, computational modeling, and mathematical analysis, they conclude that enhanced LTD can lead to both an increase and a decrease of the VOR depending on the history of synaptic changes and the details of the synaptic model.

*Reviewer #2:*

I feel the authors have largely done a good job in revising the manuscript.

The discussion of time scales has not become as transparent in the main text as I initially hoped, but the treatment in the Appendix is fine.

I'm still not fully satisfied the new statement "Whereas the plasticity stability dilemma is a memory deficit with enhanced plasticity, what we are reporting is a learning deficit." In fact people know about "Learning" problems related to too high learning rates in neural networks since half a century. E.g. see any textbook on the perceptron rule or gradient decent rule etc.

Reviewer #3:

I am generally satisfied by the revisions made by the authors.

However, I am somewhat disappointed by the answer to the first issue

(whether baseline EPSCs have different amplitudes in WT or KO).

I would have imagined that in such experiments stimulation strengths are recorded by the experimentalist. But of course I have never done such experiments, so I am not really qualified to make such a statement.

---

## [Author Response]

Essential revisions:

– The different models used by the authors assume that LTD transition probabilities are increased in KO mice compared to WT, but that LTP transition probabilities are unchanged. As a result, baseline synaptic strengths should be lower in KO than in WT. Some of these authors have presented data in the McConnell et al. 2009 paper, where the amplitude of baseline EPSCs is shown in Figure 3, but it is not fully clear there whether they are identical in WT and KO.

If they are identical, then this data might be consistent with a model in which the LTP transition rates are also increased, so that the baseline synaptic strengths are identical in both types of mice. Does such a model reproduce the data? Alternatively, if the McConnell data show a difference in WT and KO amplitudes, it would be interesting to directly compare this to the model.

The experiments described in McConnell et al. 2009 do not provide information about the basal synaptic strengths of parallel fiber input onto Purkinje cells in DKO and WT, because they adjusted the parallel fiber stimulation strength to obtain similar PF EPSC amplitudes before the LTD induction protocol, which is a common practice in slice physiology experiments. [McConnell et al. 2009, Materials and methods, Electrophysiology: “Parallel fibers were stimulated in the molecular layer…. Stimulus strength was adjusted so that the first EPSC was approximately 100 to 300 pA.”].

As suggested by the reviewers, we ran additional simulations with models in which the LTP transition rates are also increased so that the baseline synaptic strengths are identical in both types of mice. We tested this for all of the different classes of models, including the serial, binary, multistate, pooled resource, cascade, and non-uniform multistate models. None of them could reproduce the data if the LTP transition rate was increased along with the LTD rate. We include these new model results in the Supplementary Material (Appendix; Figure 11).

– Generally the parameter space of the models is not very well analyzed. So the reader is left with the impression that parameters are chosen such that the effect can or cannot be reproduced. Particularly for the model classes that do not allow analytical arguments a more comprehensive parameter check is necessary, or the choice of the parameters has to justified more convincingly.

For the serial, binary and multistate models, we show analytically that these models cannot fit the data for any parameters (Appendix; see Figure 6; Figure 7).

For the pooled resource model, we scanned all six relevant parameters and found no solutions that reproduced the data (Appendix section “Multistate model”; Figure 8).

For the models that can fit the data, we clarify that our modeling results are more of an existence proof—we are not claiming that these models will fit the data for all sets of parameters, but rather that these models can work, with an appropriate choice of parameters (Appendix section “Multistate model”, see Figure 6 ).

Thus, the key distinction we are drawing is between classes of models that can work, with appropriate choice of parameters, versus those that cannot work for any choice of parameters. We have rewritten the text to clarify these points (subsection “Modeling predicts strong saturation of LTD and difficult to reverse synaptic states”).

– Concerning the optogenetic experiments: If the optogenetic experiments really provide a test of the ideas regarding cascade models, then this experiment should be explicitly modeled using the various different plasticity models discussed. If the optogenetic results are also predicted by standard/simpler models then the discussion of this experiment and its significance should be modified, i.e. toned down, accordingly.

We now include simulations of the optogenetic experiments for each class of model (Appendix subsection “Cascade and non-uniform multistate models”; Figure 10). All models are able to reproduce the impaired learning observed in WT mice after artificial LTD induction, even the ones that fail to reproduce our other empirical observations.

The optogenetic experiments were not meant as a test of the cascade models, rather the modeling and the optogenetic experiments each provide a test the plausibility of the general idea that saturation could cause impaired learning with enhanced plasticity. In the revised text, we clarify this point, by introducing the saturation hypothesis earlier in the manuscript (Results section). In addition, we have rearranged the order in which the results are presented, with all of the experimental results (Figure 1–Figure 3) now coming before the modeling (Figure 4). This rearrangement should help to clarify that the stimulation experiments are not meant as a test of the models.

– Are other cell types in cerebellum or vestibular system affected by the mutation? If so this should be mentioned and the implications discussed.

Expression of MHC H2-K^b^ and H2-D^b^ is enriched in the Purkinje cells but is not exclusive to the Purkinje cells, and may be present in other cells contributing to the VOR behavior, as we now acknowledge in the manuscript (Results section). To address this concern about cell type specificity, we conducted a virally mediated, Purkinje cell-specific rescue of H2-D^b^ in adult DKO animals (Figure 1 hashed bars). Purkinje cell-specific expression of H2-D^b^ reversed the learning phenotype in the DKO mice, demonstrating that the learning phenotype can be attributed to the loss of H2-D^b^ expression in Purkinje cells, the post-synaptic site of pf-Pk LTD.

– There are a number of straightforward experimental predictions that naturally follow from the model that are not mentioned by the authors. One straightforward prediction is that with enough VOR-increase training, the advantage of the KO mice should eventually disappear, and in fact the WT should again become better, since the memory of initial conditions should eventually decay away. I am not suggesting the authors perform the corresponding experiments, but I believe it would be good to clarify this issue in simulations.

This is indeed a prediction of the models, as confirmed in additional simulations regarding the effects of extended training after pre-training, whose results are now included (subsection “Modeling predicts strong saturation of LTD and difficult to reverse synaptic states”; Appendix; Figure 9).

Related to that, in the Discussion section, the authors write “Within a given training session, VOR learning appears to asymptote within a few tens of minutes”. This does not seem to be true in the experiments reported in Figure 2.

We agree with the reviewer that the learning of the WT mice shown in Figure 3, left panel (original Figure 2A) does not appear to asymptote “within a few tens of minutes”. This point is not essential to our arguments, therefore we have removed this statement from the discussion.

These experiments leave open the possibility that these effects are only transient, and that given enough time WT and KO would reach the same level of learning.

The experiments in Figure 3, left panel, illustrate that during the last 10 minutes of training, the slope of learning for WT is clearly steeper than DKO, hence, there is nothing in these experiments to suggest that the DKO will catch up and reach the same level of learning as WT without pre-training.

Finally, the issue of time scales needs more discussion. For instance, on longer time scales already the 2 state model WT can overtake the KO (the top one in Figure 3B).

A discussion of time scales is now included (Results section, Appendix). It is indeed the case that over longer time scales, the total change in learning will be larger for WT than DKO and eventually there is a crossover, even in the two-state model.

*– The rates of activity in granule cells in the flocculus* in vivo *seems important for the modeling but has never been directly measured. Granule cells might be expected to fire very little spontaneously. What rates were assumed in the model? How do conclusions of the model depend on assumed activity levels in granule cells?*

The activity of granule cells in the flocculus has not been directly measured. It is known that the vestibular mossy fibers providing input to the granule cells in the flocculus have a fairly high spontaneous firing rate, so we would not necessarily expect that the granule cells in this part of the cerebellum to have low spontaneous rates. Fortunately, the conclusions of the model do not depend on assumptions about the rate of granule cells activity, as now indicated on the first page of the Appendix.

– A previous study by McConnell et al. found that rotorod learning and retention was enhanced in the same mutant mice. This seems at odds with the present findings. The authors should discuss possible explanations based on their work.

The enhanced learning reported by McConnell et al. is consistent with our conclusion that enhanced plasticity can support either enhanced or impaired learning, depending on the history of activity. Rotorod learning depends on a different part of the cerebellum than VOR learning. In the Discussion section of the revised text, we point out that different parts of the cerebellum have different levels of spontaneous activity, and hence may have different basal levels of LTD-eligible synapses, and therefore either enhanced or impaired learning in the DKOs.

– The Abstract and Introduction should be rewritten to deal more fairly with the previous literature. While the combination of techniques is novel, many elements are not. The manuscript falls short of acknowledging the fact that many researches since several decades have been studying these types problems from a variety of directions sometimes termed "plasticity stability dilemma". Just to name a few of these approaches:

a) From the theoretical perspective palimpsest models (Amit and Fusi, 1994; and many others) have very early laid out early on that an increase in learning rate also increases forgetting. b) Studies on trafficking of receptor molecules come to similar conclusions (reviewed in Gerrow and Triller, 2010) c) Papers on synaptic metaplasticity and tagging/consolidation raising similar problems

We have revised the manuscript (Introduction) to acknowledge previous work on the “plasticity stability dilemma.” We also clarify that whereas this previous work and our current work both consider disadvantages of enhanced plasticity, there is also a key difference. Whereas the plasticity stability dilemma is a memory deficit with enhanced plasticity, what we are reporting is a learning deficit.

[Editor’s note: Further clarifications were requested on acceptance. The authors’ response follows.]

Reviewer #2:

*I feel the authors have largely done a good job in revising the manuscript.*

The discussion of time scales has not become as transparent in the main text as I initially hoped, but the treatment in the Appendix is fine.

I'm still not fully satisfied the new statement "Whereas the plasticity stability dilemma is a memory deficit with enhanced plasticity, what we are reporting is a learning deficit." In fact people know about "Learning" problems related to too high learning rates in neural networks since half a century. E.g. see any textbook on the perceptron rule or gradient decent rule etc.

It is true, as the reviewer notes, that learning rates that are too fast can cause learning deficits in artificial neural networks, as is well known. However, the primary manifestation of this learning deficit is an oscillation that arises from the synaptic weights overshooting their optimal values and then returning, thereby bouncing back and forth near the vicinity of a local minimum in the error function over synaptic weight space. Furthermore, if the learning rate is extremely fast, the error can diverge. Since we see no such oscillations or divergence in our performance measure, i.e. the VOR gain, it is unlikely that such theoretical considerations in artificial networks are likely to be relevant to our experimental findings.

The reviewer is worried about a problem that is not relevant to us: fast learning rates in neural network training can impede training due to overshooting the correct set of weights - thereby yielding oscillations. We do not see oscillations and so we are simply not in that regime, thus we have decided not to modify our manuscript to include this discussion point.

Reviewer #3:

I am generally satisfied by the revisions made by the authors.

*However, I am somewhat disappointed by the answer to the first issue*

*(whether baseline EPSCs have different amplitudes in WT or KO).*

I would have imagined that in such experiments stimulation strengths are recorded by the experimentalist. But of course I have never done such experiments, so I am not really qualified to make such a statement.

In a slice experiment, the baseline EPSC amplitude is highly dependent on the exact placement of the stimulating electrode, which makes it very difficult to compare basal synaptic strengths across animals. It is not impossible, and, indeed, with the appropriate, careful design of experiments for this specific purpose, some studies have compared basal EPSC strengths across slices. However, the experiments reported in McConnell et al 2009 were not designed to compare basal EPSC strength, hence we do not think it is appropriate to use the data for that purpose.